# Discrete Probabilistic Inference as Control in Multi-path Environments

**Tristan Deleu**[1,3]    **Padideh Nouri**[2]    **Nikolay Malkin**[1]    **Doina Precup**[2,4]    **Yoshua Bengio**[1]

Mila – Quebec AI Institute

[1]Université de Montréal   [2]McGill University   [3]Valence Labs   [4]Google DeepMind

## Abstract

We consider the problem of sampling from a discrete and structured distribution as a sequential decision problem, where the objective is to find a stochastic policy such that objects are sampled at the end of this sequential process proportionally to some predefined reward. While we could use maximum entropy Reinforcement Learning (MaxEnt RL) to solve this problem for some distributions, it has been shown that in general, the distribution over states induced by the optimal policy may be biased in cases where there are multiple ways to generate the same object. To address this issue, Generative Flow Networks (GFlowNets) learn a stochastic policy that samples objects proportionally to their reward by approximately enforcing a conservation of flows across the whole Markov Decision Process (MDP). In this paper, we extend recent methods correcting the reward in order to guarantee that the marginal distribution induced by the optimal MaxEnt RL policy is proportional to the original reward, regardless of the structure of the underlying MDP. We also prove that some flow-matching objectives found in the GFlowNet literature are in fact equivalent to well-established MaxEnt RL algorithms with a corrected reward. Finally, we study empirically the performance of multiple MaxEnt RL and GFlowNet algorithms on multiple problems involving sampling from discrete distributions.

## 1   INTRODUCTION

Approximate probabilistic inference has seen a tremendous amount of progress, notably from a variational perspective

Correspondence to: Tristan Deleu <deleutri@mila.quebec>
Code: https://github.com/tristandeleu/gfn-maxent-rl

coupled with deep neural networks. This is particularly true for continuous sample spaces, where the Evidence Lower Bound (ELBO) can be maximized with gradient methods, thanks to methods such as pathwise gradient estimation [Kingma and Welling, 2014, Rezende et al., 2014, Rezende and Mohamed, 2015]. In the case of discrete and highly structured sample spaces though, this "reparametrization trick" becomes more challenging since it often requires continuous relaxations of discrete distributions [Jang et al., 2017, Maddison et al., 2017, Mena et al., 2018]. In those cases, variational inference can also be carried out more generally thanks to score function estimation [Williams, 1992], albeit at the expense of high variance [Mohamed et al., 2020]. An alternative for approximate inference is through sampling methods, based on Markov chain Monte Carlo (MCMC; Hastings, 1970, Gelfand and Smith, 1990), when the target distribution is defined up to an intractable normalization constant.

Bengio et al. [2021] introduced a new class of probabilistic models called *Generative Flow Networks* (GFlowNets), to approximate an unnormalized target distribution over discrete and structured sample spaces from a variational perspective [Malkin et al., 2023, Zimmermann et al., 2022]. GFlowNets treat sampling as a sequential decision making problem, heavily inspired by the literature in Reinforcement Learning (RL). Unlike RL though, which seeks an optimal policy maximizing the cumulative reward, the objective of a GFlowNet is to find a policy such that objects can be sampled proportionally to their cumulative reward. Nevertheless, this relationship led to a number of best practices from the RL literature being transferred into GFlowNets, such as the use of a replay buffer [Shen et al., 2023, Vemgal et al., 2023] and target network [Deleu et al., 2022], and advanced exploration strategies [Rector-Brooks et al., 2023].

Although Bengio et al. [2021] proved that GFlowNets were exactly equivalent to maximum entropy RL (MaxEnt RL; Ziebart, 2010) in some specific cases which we will recall in Section 2, it has long been thought that this connection was only superficial in general. However recently, Tiapkin et al.

[2024] showed that GFlowNets and MaxEnt RL are in fact one and the same, up to a correction of the reward function. This, along with other recent works [Mohammadpour et al., 2024, Niu et al., 2024], paved the way to show deeper connections between GFlowNets and MaxEnt RL algorithms. In this work, we extend this correction of the reward to a more general case, and establish novel equivalences between GFlowNet & MaxEnt RL objectives, notably between the widely used Trajectory Balance loss in the GFlowNet literature [Malkin et al., 2022], and the Path Consistency Learning algorithm in MaxEnt RL [Nachum et al., 2017]. We also introduce a variant of the Soft Q-Learning algorithm [Haarnoja et al., 2017], depending directly on a policy, and show that it becomes equivalent to the Modified Detailed Balance loss of Deleu et al. [2022]. Finally, we show these similarities in behavior empirically on three different domains, and include the popular Soft Actor-Critic algorithm (SAC; Haarnoja et al., 2018a) in our evaluations, which has no existing GFlowNet counterpart, under similar conditions on the same domains.

## 2 MARGINAL SAMPLING VIA SEQUENTIAL DECISION MAKING

Given an energy function $\mathcal{E}(x)$ defined over a discrete and finite sample space $\mathcal{X}$, our objective is to sample objects $x \in \mathcal{X}$ from the *Gibbs distribution*:

$$P(x) \propto \exp(-\mathcal{E}(x)/\alpha), \qquad (1)$$

where $\alpha > 0$ is a temperature parameter. We assume that this energy function is fixed and we can query it for any element in $\mathcal{X}$. Sampling from this distribution is in general a challenging problem due to the partition function $Z = \sum_{x \in \mathcal{X}} \exp(-\mathcal{E}(x)/\alpha)$ acting as the normalization constant for $P$, which is intractable when the sample space is (combinatorially) large. Throughout this paper, we will focus on cases where the objects of interest have some compositional structure (*e.g.*, graphs, or trees), meaning that they can be constructed piece by piece.

### 2.1 MAXIMUM ENTROPY REINFORCEMENT LEARNING

We consider a finite-horizon Markov Decision Process (MDP) $\mathcal{M}_{\text{soft}} = (\mathcal{S}, \mathcal{A}, s_0, T, r)$, where the state space $\mathcal{S}$ and the action space $\mathcal{A}$ are discrete and finite. We assume that this MDP is deterministic, with a transition function $T : \mathcal{S} \times \mathcal{A} \to \bar{\mathcal{S}}$ that determines how to move to a new state $s' = T(s, a)$ from the state $s$, following the action $a$. We identify an initial state $s_0 \in \mathcal{S}$ from which all the trajectories start. Moreover, since we are in a finite-horizon setting, all these trajectories are finite and we assume that they eventually end at an abstract terminal state $s_f \notin \mathcal{S}$ acting as a "sink" state; we use the notation $\bar{\mathcal{S}} = \mathcal{S} \cup \{s_f\}$.

The state space is defined as a superset of the sample space $\mathcal{X} \subseteq \mathcal{S}$, and is structured in such a way that $x \in \mathcal{X}$ iff we can transition from $x$ to the terminal state $s_f$ (*i.e.*, there exists an action $a \in \mathcal{A}$ such that $T(x, a) = s_f$); the states $x \in \mathcal{X}$ are called *terminating states*, following the naming convention of Bengio et al. [2023]. We also set the discount factor $\gamma = 1$ throughout this paper. This setting with known deterministic dynamics is well-studied in the Reinforcement Learning literature [Todorov, 2006, Kappen et al., 2012].

Since the MDP is deterministic, we can identify the action $a$ leading to a state $s' = T(s, a)$ with the transition $s \to s'$ in the state space itself. As such, we will write all quantities involving state-action pairs $(s, a)$ in terms of $(s, s')$ instead. The reward function $r(s, s')$ is defined such that the sum of rewards along a complete trajectory (the return) only depends on the energy of the terminating state it reaches: for a trajectory $\tau = (s_0, s_1, \ldots, s_T, s_f)$, we have

$$\sum_{t=0}^{T} r(s_t, s_{t+1}) = -\mathcal{E}(s_T), \qquad (2)$$

with the convention $s_{T+1} = s_f$. In particular, this covers the case of a sparse reward that is received only at the end of the trajectory (*i.e.*, $r(s_T, s_f) = -\mathcal{E}(s_T)$, and zero everywhere else). This decomposition of the energy into intermediate rewards is similar to Buesing et al. [2020].

While the objective of Reinforcement Learning is typically to find a policy $\pi(s_{t+1} \mid s_t)$ maximizing the expected sum of rewards (here corresponding to finding a state $x \in \mathcal{X}$ with lowest energy, or equivalently a mode of $P$ in (1)), in *maximum entropy Reinforcement Learning* (MaxEnt RL) we also search for a policy that maximizes the expected sum of rewards, but this time augmented with the entropy $\mathcal{H}(\pi(\cdot \mid s))$ of the policy $\pi$ in state $s$:

$$\pi_{\text{MaxEnt}}^{*} = \arg\max_{\pi} \mathbb{E}_{\tau}\left[\sum_{t=0}^{T} r(s_t, s_{t+1}) + \alpha\mathcal{H}(\pi(\cdot \mid s_t))\right], \qquad (3)$$

where the expectation is taken over complete trajectories $\tau$ sampled following the policy $\pi$. Intuitively, adding the entropy to the objective encourages stochasticity in the optimal policy and improves diversity. To highlight the difference with the standard objective in RL, $\mathcal{M}_{\text{soft}}$ will be called *soft MDP*, following the nomenclature of Ziebart [2010].

**Notations.** In what follows, it will be convenient to view the transitions in the soft MDP as a directed acyclic graph (DAG) $\mathcal{G}$ over the states in $\bar{\mathcal{S}}$ (including $s_f$), rooted in $s_0$; the graph is acyclic to ensure that $\mathcal{M}_{\text{soft}}$ is finite-horizon. We will use the notations $\text{Pa}(s)$ and $\text{Ch}(s)$ to denote respectively the parents and the children of a state $s$ in $\mathcal{G}$. For any terminating state $x \in \mathcal{X}$, $s_0 \rightsquigarrow x$ denotes a complete trajectory in $\mathcal{G}$ of the form $\tau = (s_0, s_1, \ldots, x, s_f)$; the tran-

sition to the terminal state is implicit in the notation, albeit necessary in our definition.

## 2.2 SAMPLING TERMINATING STATES FROM A SOFT MDP

From the literature on *control as inference* [Ziebart et al., 2008, Levine, 2018], it can be shown that the policy maximizing the MaxEnt RL objective in (3) induces a distribution over dynamically consistent trajectories $\tau = (s_0, s_1, \ldots, s_T, s_f)$ that depends on the sum of rewards along the trajectory:

$$\pi^*(\tau) \triangleq \prod_{t=0}^{T} \pi^*_{\text{MaxEnt}}(s_{t+1} \mid s_t) \propto \exp\left(\frac{1}{\alpha}\sum_{t=0}^{T} r(s_t, s_{t+1})\right). \tag{4}$$

With our choice of reward function in (2), this suggests a simple strategy to sample from the Gibbs distribution in (1), once the optimal policy $\pi^*_{\text{MaxEnt}}$ is known: sample a trajectory in $\mathcal{M}_{\text{soft}}$, starting at the initial state $s_0$ and following the optimal policy, and only return the terminating state $s_T$ reached at the end of $\tau$ (ignoring the terminal state $s_f$). However, in general, we are interested in sampling an object $x \in \mathcal{X}$ (the terminating state) using this sequential process, but not *how* this object is generated (the exact trajectory taken). The distribution of interest is therefore not a distribution over *trajectories* as in (4), but its *marginal* over terminating states:

$$\pi^*(x) \triangleq \sum_{\tau: s_0 \rightsquigarrow x} \pi^*(\tau). \tag{5}$$

$\pi^*(x)$ is called the *terminating state distribution* associated with the policy $\pi^*_{\text{MaxEnt}}$ [Bengio et al., 2023]. When the state transition graph $\mathcal{G}$ of the soft MDP is a tree[1] rooted in $s_0$, and there is a unique complete trajectory $s_0 \rightsquigarrow x$ leading to any $x \in \mathcal{X}$, then this process is guaranteed to be equivalent to sampling from the Gibbs distribution [Bengio et al., 2021]. Examples of tree-structured MDPs include the autoregressive generation of sequences [Bachman and Precup, 2015, Weber et al., 2015, Angermueller et al., 2020, Jain et al., 2022, Feng et al., 2022], and sampling from a discrete factor graph with a fixed ordering of the random variables [Buesing et al., 2020].

The equivalence between control and inference (sampling from (1)) no longer holds when $\mathcal{G}$ is a general DAG though. As an illustrative example, shown in Figure 1, consider the problem of generating small molecules by adding fragments one at a time, as described in You et al. [2018]. If we follow the optimal policy, even though the distribution over trajectories is proportional to $\exp(-\mathcal{E}(x))$ as in (4), its marginal

---

[1]With the exception of the terminal state $s_f$, whose parents are always all the terminating states in $\mathcal{X}$.

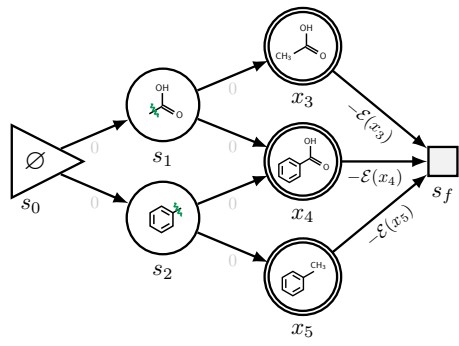

| $x$ | $\tau$ | $\pi^*(\tau)$ | $\pi^*(x)$ |
|---|---|---|---|
| $x_3$ | $s_0 \to s_1 \to x_3$ | $\exp(-\mathcal{E}(x_3))/Z'$ | $\exp(-\mathcal{E}(x_3))/Z'$ |
| $x_4$ | $s_0 \to s_1 \to x_4$ | $\exp(-\mathcal{E}(x_4))/Z'$ | $2\exp(-\mathcal{E}(x_4))/Z'$ |
| | $s_0 \to s_2 \to x_4$ | $\exp(-\mathcal{E}(x_4))/Z'$ | |
| $x_5$ | $s_0 \to s_2 \to x_5$ | $\exp(-\mathcal{E}(x_5))/Z'$ | $\exp(-\mathcal{E}(x_5))/Z'$ |

Figure 1: Illustration of the bias of the terminating state distribution associated with $\pi^*_{\text{MaxEnt}}$ on a soft MDP with a DAG structure. The labels on each transition of the MDP corresponds to the reward function, satisfying (2) (sparse reward setting). The terminating state distribution $\pi^*(x)$ is computed by marginalizing $\pi^*(\tau)$ over trajectories leading to $x$ (*e.g.*, two trajectories $s_0 \to s_1 \to x_4$ and $s_0 \to s_2 \to x_4$ to $x_4$). $\pi^*(\tau)$ is computed based on (4), and we assume $\alpha = 1$. The terminating state distribution $\pi^*(x)$ should be contrasted with the (target) Gibbs distribution $P(x) \propto \exp(-\mathcal{E}(x))$. The normalization constant is $Z' = \exp(-\mathcal{E}(x_3)) + 2\exp(-\mathcal{E}(x_4)) + \exp(-\mathcal{E}(x_5))$. This MDP is inspired by Jain et al. [2023a].

over terminating states is biased since there are two trajectories leading to the same molecule $x_4$ (*i.e.*, multiple orders in which fragments can be added). This bias was first highlighted by Bengio et al. [2021].

## 2.3 GENERATIVE FLOW NETWORKS

To address this mismatch between the target Gibbs distribution in (1) and the terminating state distribution in (5) induced by the optimal policy in MaxEnt RL, Bengio et al. [2021] introduced a new class of probabilistic models over discrete and compositional objects called *Generative Flow Networks* (GFlowNets; Bengio et al., 2023). Instead of searching for a policy maximizing (3), the goal of a GFlowNet is to find a *flow* function $F(s \to s')$ defined over the edges of $\mathcal{G}$ that satisfies the following flow-matching conditions for all states $s' \in \mathcal{S}$ such that $s' \neq s_0$:

$$\sum_{s \in \text{Pa}(s')} F(s \to s') = \sum_{s'' \in \text{Ch}(s')} F(s' \to s''), \tag{6}$$

with an additional boundary condition for each terminating state $x \in \mathcal{X}$: $F(x \to s_f) = \exp(-\mathcal{E}(x)/\alpha)$. Putting it in

words, (6) means that the total amount of flow going into any state $s'$ has to be equal to the amount of flow going out of it, except for the initial state $s_0$ which acts as a single "source" for all the flow. We can define a policy[2] from a flow function by simply normalizing the flows going out of any state $s_t$:

$$P_F(s_{t+1} \mid s_t) \propto F(s_t \to s_{t+1}). \quad (7)$$

Bengio et al. [2021] showed that if there exists a flow function satisfying the flow-matching conditions in (6) as well as the boundary conditions, then sampling terminating states by following $P_F$ in the soft MDP as described in Section 2.2 is equivalent to sampling from (1). In that case, the terminating state distribution associated with the policy $P_F$ satisfies $P_F(x) \propto \exp(-\mathcal{E}(x)/\alpha)$, for any terminating state $x \in \mathcal{X}$, and is not biased by the number of trajectory leading to $x$, contrary to the optimal policy in MaxEnt RL.

# 3 BRIDGING THE GAP BETWEEN MAXENT RL & GFLOWNETS

There exists a fundamental difference between MaxEnt RL and GFlowNets in the way the distributions induced by their (optimal) policies relate to the energy: in MaxEnt RL, this distribution is over *how* objects are being created (trajectories) as mentioned in Section 2.2, whereas a GFlowNet induces a distribution over the *outcomes* only (terminating states), the latter matching the requirements of the Gibbs distribution. In this section, we recall some existing connections between GFlowNets and MaxEnt RL [Tiapkin et al., 2024], and we establish new equivalences between existing MaxEnt RL algorithms and GFlowNet objectives.

## 3.1 REWARD CORRECTION

To correct the bias illustrated in Figure 1 caused by multiple trajectories leading to the same terminating state, we can treat (5) not as a sum but as an *expectation* over trajectories, by reweigthing $\pi^*(\tau)$ (which is constant for a fixed terminating state, by (4) & (2)) with a probability distribution over these trajectories. Bengio et al. [2023] showed that such a distribution over complete trajectories can be defined by introducing a *backward transition probability* $P_B(s \mid s')$, which is a distribution over the parents $s \in \mathrm{Pa}(s')$ of any state $s' \neq s_0$. Tiapkin et al. [2024] showed that the reward of the soft MDP can be modified based on $P_B$ in such a way that the corresponding optimal policy $\pi^*_{\mathrm{MaxEnt}}$ is equal to the GFlowNet policy $P_F$ in (7). We restate and generalize this result in Theorem 3.1, where we show how this correction counteracts the effect of the marginalization in (5),

---

[2]Following the conventions from the GFlowNet literature, we will use the notation $P_F$ for this policy, also called the forward transition probability [Bengio et al., 2023], to distinguish it from the optimal policy $\pi^*_{\mathrm{MaxEnt}}$ in MaxEnt RL.

resulting in a terminating state distribution that matches the Gibbs distribution.

**Theorem 3.1** (Gen. of Tiapkin et al., 2024; Theorem 1). *Let $P_B(\cdot \mid s')$ be an arbitrary backward transition probability (i.e., a distribution over the parents of $s' \neq s_0$ in $\mathcal{G}$). Let $r(s, s')$ be the reward function of the MDP corrected with $P_B$, satisfying for any trajectory $\tau = (s_0, s_1, \ldots, s_T, s_f)$:*

$$\sum_{t=0}^{T} r(s_t, s_{t+1}) = -\mathcal{E}(s_T) + \alpha \sum_{t=0}^{T-1} \log P_B(s_t \mid s_{t+1}), \quad (8)$$

*where we used the convention $s_{T+1} = s_f$. Then the terminating state distribution associated with the optimal policy $\pi^*_{\mathrm{MaxEnt}}$ solution of (3) satisfies $\pi^*(x) \propto \exp(-\mathcal{E}(x)/\alpha)$.*

The proof of the theorem is available in Appendix A. Unlike (2), the return now depends on the trajectory leading to $s_T$ via the second term in (8). Interestingly, the temperature parameter $\alpha$ introduced in the MaxEnt RL literature [Haarnoja et al., 2017] finds a natural interpretation as the temperature of the Gibbs distribution. Note that the correction in (8) only involves the backward probability of the whole trajectory $\tau$, making it also compatible even with non-Markovian $P_B$ [Shen et al., 2023, Bengio et al., 2023].

Tiapkin et al. [2024] only considered the case where the reward function of the soft MDP is sparse, and the correction with $P_B(s_t \mid s_{t+1})$ is added at each intermediate transition $s_t \to s_{t+1}$; we will go back to this setting in the following section. This correction of the reward is fully compatible with our observation in Section 2.2 that sampling terminating states with $\pi^*_{\mathrm{MaxEnt}}$ yields samples of (1) when the soft MDP is a tree with the (uncorrected) reward in (2), since in that case any state $s' \neq s_0$ has a unique parent $s$, and thus $P_B(s \mid s') = 1$ as also observed by Tiapkin et al. [2024].

## 3.2 EQUIVALENCE BETWEEN PCL & (SUB)TB

Similar to how (2) covered the particular case of a sparse reward, in this section we will consider a reward function satisfying (8) where the energy function only appears at the end of the trajectory

$$\begin{aligned} r(s_t, s_{t+1}) &= \alpha \log P_B(s_t \mid s_{t+1}) \\ r(s_T, s_f) &= -\mathcal{E}(s_T), \end{aligned} \quad (9)$$

as introduced in [Tiapkin et al., 2024]. Theorem 3.1 suggests that solving the MaxEnt RL problem in (3) with the corrected reward is comparable to finding a solution of a GFlowNet, as they both lead to a policy whose terminating state distribution is the Gibbs distribution. It turns out that there exists an equivalence between specific algorithms solving these two problems with our choice of reward function above: Path Consistency Learning (PCL; Nachum et al., 2017) for MaxEnt RL, and the Subtrajectory Balance objective (SubTB; Madan et al., 2023, Malkin et al., 2022) for

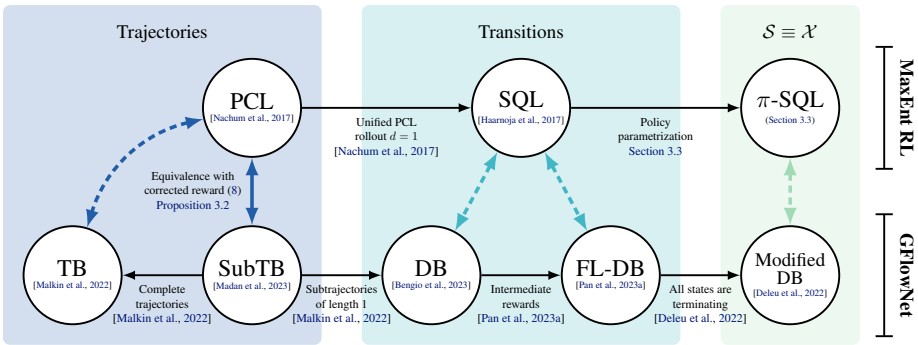

Figure 2: Equivalence between objectives in MaxEnt RL, with corrected rewards, and the objectives in GFlowNets. The objectives are classified based on whether they operate at the level of (complete) trajectories (left), transitions (middle), or if all the states are terminating (right). Further details about the form of the different residuals and the correspondences to transfer from one objective to another are available in Figure 6.

GFlowNets. Both of these objectives operate at the level of partial trajectories of the form $\tau = (s_m, s_{m+1}, \ldots, s_n)$, where $s_m$ and $s_n$ are not necessarily the initial and terminal states anymore.

On the one hand, the PCL objective $\mathcal{L}_{\text{PCL}}(\theta, \phi) = \frac{1}{2}\mathbb{E}_{\pi_b}[\Delta^2_{\text{PCL}}(\tau; \theta, \phi)]$ encourages the consistency between a policy $\pi_\theta$ parametrized by $\theta$ and a soft value function $V^\phi_{\text{soft}}$ parametrized by $\phi$, where $\pi_b$ is an arbitrary distribution over (partial) trajectories $\tau$, and the residual is defined as

$$\Delta_{\text{PCL}}(\tau; \theta, \phi) = -V^\phi_{\text{soft}}(s_m) + V^\phi_{\text{soft}}(s_n) \qquad (10)$$
$$+ \sum_{t=m}^{n-1} \big(r(s_t, s_{t+1}) - \alpha \log \pi_\theta(s_{t+1} \mid s_t)\big).$$

On the other hand, the SubTB objective $\mathcal{L}_{\text{SubTB}}(\theta, \phi) = \frac{1}{2}\mathbb{E}_{\pi_b}[\Delta^2_{\text{SubTB}}(\tau; \theta, \phi)]$ also enforces some form of consistency, but this time between a policy (forward transition probability) $P^\theta_F$ parametrized by $\theta$, and a state flow function $F_\phi$ parametrized by $\phi$, and the residual is defined as

$$\Delta_{\text{SubTB}}(\tau; \theta, \phi) = \log \frac{F_\phi(s_n) \prod_{t=m}^{n-1} P_B(s_t \mid s_{t+1})}{F_\phi(s_m) \prod_{t=m}^{n-1} P^\theta_F(s_{t+1} \mid s_t)},$$

where $P_B$ is a backward transition probability, which we assume to be fixed here—although in general, $P_B$ may also be learned [Malkin et al., 2022]. In addition to this objective, some boundary conditions on $F_\phi$ must also be enforced (depending on $\mathcal{E}$), similar to the ones introduced in Section 2.3; see Appendix B.1 for details. The following proposition establishes the equivalence between these two objectives, up to a normalization constant that only depends on the temperature $\alpha$, and provides a way to move from the policy/value function parametrization in MaxEnt RL to the policy/flow function parametrization in GFlowNets. Although similarities between these two methods have been mentioned in prior work [Malkin et al., 2022, Jiralerspong et al., 2024, Hu et al., 2024, Mohammadpour et al., 2024], we show here an exact equivalence between both objectives.

**Proposition 3.2.** *The Subtrajectory Balance objective (GFlowNet; Madan et al., 2023) is proportional to the Path Consistency Learning objective (MaxEnt RL; Nachum et al., 2017) on the soft MDP with the reward function defined in (9), in the sense that $\mathcal{L}_{\text{PCL}}(\theta, \phi) = \alpha^2 \mathcal{L}_{\text{SubTB}}(\theta, \phi)$, with the following correspondence*

$$\pi_\theta(s' \mid s) = P^\theta_F(s' \mid s) \qquad V^\phi_{\text{soft}}(s) = \alpha \log F_\phi(s). \quad (11)$$

The proof of this proposition is available in Appendix B.1. The equivalence between the value function in MaxEnt RL and the state flow function in GFlowNets was also found in [Tiapkin et al., 2024]. When applied to complete trajectories, this also shows the connection between the Trajectory Balance objective (TB; Malkin et al., 2022), widely used in the GFlowNet literature, and PCL under our choice of corrected reward in (9).

On the other end of the spectrum, if we apply this proposition to transitions in the soft MDP (*i.e.*, subtrajectories of length 1), then we can obtain a similar equivalence between the Detailed Balance objective in GFlowNets (DB; Bengio et al., 2023), and the Soft Q-Learning algorithm (SQL; Haarnoja et al., 2017), via the *Unified PCL* perspective of Nachum et al. [2017] that uses a soft Q-function in order to simultaneously parametrize both the policy and the value function in (10); see Corollary B.1 for a detailed statement. We note that this connection between DB & SQL was also mentioned in prior work [Tiapkin et al., 2024, Mohammadpour et al., 2024]. A full summary of the connections resulting from Proposition 3.2 between different MaxEnt RL and GFlowNet objectives is available in Figure 2, with further details in Figure 6.

### 3.3 SOFT Q-LEARNING WITH POLICY PARAMETRIZATION

In this section, we consider the case where all the states of the soft MDP are valid elements of the sample space $\mathcal{X}$

(in other words, all the states are terminating). Since every state is now associated with some energy, we can reshape the rewards [Ng et al., 1999] while still satisfying (8) as

$$r(s_t, s_{t+1}) = \mathcal{E}(s_t) - \mathcal{E}(s_{t+1}) + \alpha \log P_B(s_t \mid s_{t+1})$$
$$r(s_T, s_f) = 0, \tag{12}$$

if we assume, without loss of generality, that $\mathcal{E}(s_0) = 0$ (any offset added to the energy function leaves (1) unchanged). This is a novel setting that differs from Tiapkin et al. [2024], and was made possible thanks to our general statement in Theorem 3.1. We show in Proposition B.2 that with our choice of rewards above, in particular the fact that no reward is received upon termination, we can express the objective of Soft Q-Learning as a function of a policy $\pi_\theta$ parametrized by $\theta$, instead of a Q-function; we call this $\pi$-*SQL*. The objective can be written as $\mathcal{L}_{\pi\text{-SQL}}(\theta) = \frac{1}{2}\mathbb{E}_{\pi_b}[\Delta^2_{\pi\text{-SQL}}(s, s'; \theta)]$, where $\pi_b$ is an arbitrary distribution over transitions $s \to s'$ such that $s' \neq s_f$, and

$$\Delta_{\pi\text{-SQL}}(s, s'; \theta) = \alpha \big[ \log \pi_\theta(s' \mid s) - \log \pi_\theta(s_f \mid s)$$
$$+ \log \pi_\theta(s_f \mid s') \big] - r(s, s'). \tag{13}$$

With the reward function in (12), this alternative perspective on SQL is remarkable in that it is equivalent to the Modified Detailed Balance objective (Modified DB; Deleu et al., 2022), specifically derived in the special case of GFlowNets whose states are all terminating. This objective can be written as $\mathcal{L}_{\text{M-DB}}(\theta) = \frac{1}{2}\mathbb{E}_{\pi_b}[\Delta^2_{\text{M-DB}}(s, s'; \theta)]$ that depends on a policy (forward transition probability) $P_F^\theta$ parametrized by $\theta$, where

$$\Delta_{\text{M-DB}}(s, s'; \theta) \tag{14}$$
$$= \log \frac{\exp(-\mathcal{E}(s')/\alpha)P_B(s \mid s')P_F^\theta(s_f \mid s)}{\exp(-\mathcal{E}(s)/\alpha)P_F^\theta(s' \mid s)P_F^\theta(s_f \mid s')}.$$

**Proposition 3.3.** *Suppose that all the states of the soft MDP are terminating $\mathcal{S} \equiv \mathcal{X}$. The Modified Detailed Balance objective (GFlowNet; Deleu et al., 2022) is proportional to the Soft Q-Learning objective with a policy parametrization (MaxEnt RL; $\pi$-SQL) on the soft MDP with the reward function defined in (12), in the sense that $\mathcal{L}_{\pi\text{-SQL}}(\theta) = \alpha^2 \mathcal{L}_{\text{M-DB}}(\theta)$, with $\pi_\theta(s' \mid s) = P_F^\theta(s' \mid s)$.*

The proof is available in Appendix B.3. This result can be further generalized to cases where the states are not necessarily all terminating, but where some partial reward can be received along the trajectory, with an equivalence between SQL and the Forward-Looking Detailed Balance objective (FL-DB; Pan et al., 2023a) in GFlowNets; see Appendix B.4 for details.

## 4  RELATED WORK

**Maximum Entropy Reinforcement Learning.**  Unlike standard reinforcement learning where an optimal policy

may be completely deterministic (at least in the fully observable case; Sutton and Barto, 2018), MaxEnt RL seeks a *stochastic* policy that balances between reward maximization and maximal entropy of future actions [Ziebart, 2010, Fox et al., 2016]. This type of entropy regularization falls into the broader domain of regularized MDPs [Geist et al., 2019]. This can be particularly beneficial for improving exploration [Haarnoja et al., 2017] and for robust control under model misspecification [Eysenbach and Levine, 2022]. Popular MaxEnt RL methods include Soft Q-Learning, Path Consistency Learning [Nachum et al., 2017], and Soft Actor-Critic [Haarnoja et al., 2018a] studied in this paper.

**Generative Flow Networks.**  Bengio et al. [2021] took inspiration from reinforcement learning and introduced GFlowNets as a solution for finding diverse molecules binding to a target protein. Since then, they have found applications in a number of domains in scientific discovery [Jain et al., 2022, 2023a, Mila AI4Science et al., 2023], leveraging diversity in conjunction with active learning [Jain et al., 2023b, Hernandez-Garcia et al., 2023], but also in combinatorial optimization [Zhang et al., 2023a,b], causal discovery [Deleu et al., 2022, 2023, Atanackovic et al., 2023], and probabilistic inference in general [Zhang et al., 2022b, Hu et al., 2023, 2024, Falet et al., 2024]. Although they were framed differently, GFlowNets are also deeply connected to the literature on variational inference [Malkin et al., 2023, Zimmermann et al., 2022].

A number of works have recently established connections between GFlowNets and (maximum entropy) RL. Closest to our work, Tiapkin et al. [2024] showed how the reward in MaxEnt RL can be corrected based on some backward transition probability to be equivalent to GFlowNets. Although their analysis is limited to the case where the reward function in the original soft MDP is sparse (*i.e.*, the reward is only obtained at the end of the trajectory), they were the first to propose a correction applied at each intermediate transition as in Section 3.2. We generalized this in Theorem 3.1 with a correction at the level of the *trajectories*, which offers more flexibility in how the correction is distributed along the trajectory and allows intermediate rewards. Tiapkin et al. [2024] also showed similarities between GFlowNet objectives and MaxEnt RL algorithms, namely between Detailed Balance & Dueling Soft Q-Learning [Wang et al., 2016], and between Trajectory Balance & Policy Gradient [Schulman et al., 2017]. The correspondence between TB and Policy Gradient was further expanded in [Niu et al., 2024], as a direct consequence of the connections between GFlowNets, variational inference [Malkin et al., 2023], and reinforcement learning [Weber et al., 2015]. Finally, Mohammadpour et al. [2024] also introduced a correction that depends on $n(s)$ the number of (partial) trajectories to a certain state $s$, which can be learned by solving a second MaxEnt RL problem (3) on an "inverse" MDP. This correction corresponds to a particular choice of backward transition probability

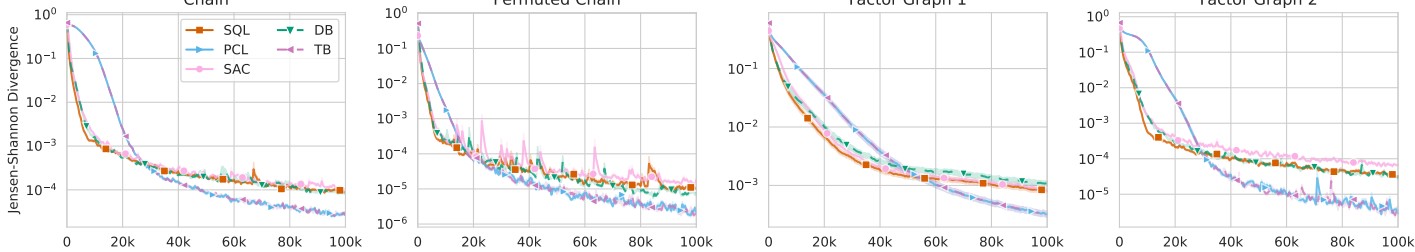

Figure 3: Comparison of MaxEnt RL and GFlowNet algorithms on the factor graph inference task, in terms of the Jensen-Shannon divergence between the terminating state distribution and the target distribution during training. Each curve represents the average JSD with 95% confidence interval over 20 random seeds.

$P_B(s_t \mid s_{t+1}) = n(s_t)/n(s_{t+1})$ in (8), which has the remarkable property to maximize the flow entropy.

## 5 EXPERIMENTAL RESULTS

We verify empirically the equivalences established in Section 3 on three domains: the inference over discrete factor graphs [Buesing et al., 2020], Bayesian structure learning of Bayesian networks [Deleu et al., 2022], and the generation of parsimonious phylogenetic trees [Zhou et al., 2024]. In addition to Detailed Balance (and possibly its modified version) and Trajectory Balance on the one hand (GFlowNets), and Soft Q-Learning (possibly parametrized by a policy; see Section 3.3) and Path Consistency Learning on the other hand (MaxEnt RL), we also consider a discrete version of Soft Actor-Critic [Christodoulou, 2019], which has no natural conterpart in the GFlowNet literature. For all MaxEnt RL methods, we adjust the MDP to include the correction of the reward. Note that in all the domains considered here, non-trivial intermediate rewards are available in the original MDP, meaning in particular that all instances of DB actually use the Forward-Looking formulation [Pan et al., 2023a]; the form of these intermediate rewards along with additional experimental details are available in Appendix C.

### 5.1 PROBABILISTIC INFERENCE OVER DISCRETE FACTOR GRAPHS

The probabilistic inference task in Buesing et al. [2020] consists in sequentially sampling the values of $d$ discrete random variables in a factor graph one at a time, with a fixed order. This makes the underlying MDP having a tree structure, eliminating the need for reward correction, as described in Section 2.2. We adapted this environment to have multiple trajectories leading to each terminating state by allowing sampling the random variables in any order. Details about the energy function are available in Appendix C.1.

In Figure 3, we show the performance of the different MaxEnt RL and GFlowNet algorithms on 4 different factor graph structures, as proposed by Buesing et al. [2020], with $d = 6$

variables and where each variable can take one of 5 possible values. We observe that TB & PCL perform similarly, validating Proposition 3.2, and overall outperform all other methods. Similarly, we can see that DB & SQL also perform similarly as expected by Corollary B.1. Finally, although SAC is generally viewed as a strong algorithm for continuous control [Haarnoja et al., 2018b], we did not observe any significant improvement over DB/SQL.

### 5.2 STRUCTURE LEARNING OF BAYESIAN NETWORKS

We also evaluated all algorithms on the task of learning the structure of Bayesian networks, using a Bayesian perspective [Deleu et al., 2022]. Given a dataset of observations $\mathcal{D}$ from a joint distribution over $d$ random variables $X_1, \ldots, X_d$, our objective is to approximate the posterior distribution $P(G \mid \mathcal{D}) \propto P(\mathcal{D} \mid G)P(G)$ over the DAG structures $G$ encoding the conditional independencies of $P(X_1, \ldots, X_d)$. The soft MDP is constructed as in [Deleu et al., 2022], where a DAG $G$ is constructed by adding one edge at a time, starting from a completely empty graph, while enforcing the acyclicity of the graph at each step of generation (*i.e.*, an edge cannot be added if it would introduce a cycle).

Following Malkin et al. [2023], we consider here a relatively small task where $d = 5$, so that the target distribution $P(G \mid \mathcal{D})$ can be evaluated analytically in order to compare it to our approximations given by MaxEnt RL and/or GFlowNets. Unlike in Section 5.1, we also included the modified DB loss and $\pi$-SQL in our comparison since all the states are valid DAGs. We observe in Figure 4 that again TB & PCL on the one hand, but also modified DB & $\pi$-SQL on the other hand perform very similarly to one another, empirically validating our equivalences established above. Despite a light search over hyperparameters, we found that SAC performs on average significantly worse than other methods, mainly due to instability during training.

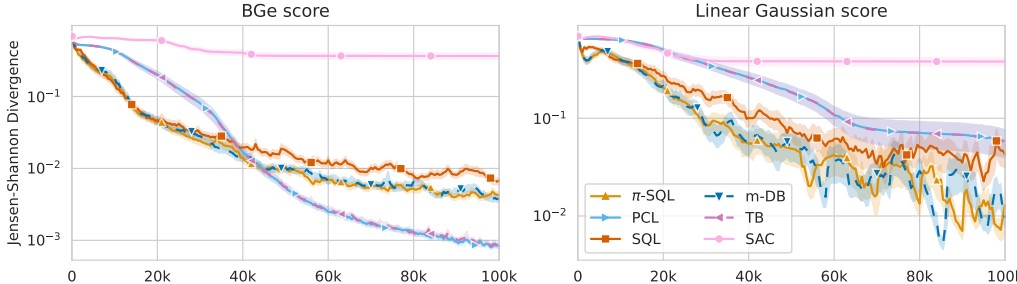

Figure 4: Comparison of MaxEnt RL and GFlowNet algorithms on the Bayesian structure learning task, in terms of the Jensen-Shannon divergence between the terminating state distribution and the target posterior during training. Both experiments differ in the way the marginal likelihood $P(\mathcal{D} \mid G)$ is computed, (left) using the BGe score [Geiger and Heckerman, 1994], (right) is the linear Gaussian score [Nishikawa-Toomey et al., 2023]. Each curve represents the average JSD with 95% confidence interval over 20 random seeds.

## 5.3 PHYLOGENETIC TREE GENERATION

Finally, we also compared these methods on the larger-scale task of parsimonious phylogenetic tree generation introduced by Zhou et al. [2024]. Based on biological sequences of different species, the objective is to find phylogenetic trees over those species that require few mutations. A state of the soft MDP corresponds to a collection of trees over a partition of the species, and actions correspond to merging two trees together by adding a root node. Note that all the trees sampled this way have the same size, although they may have different number of trajectories leading to each of them (unlike Figure 3). The energy of a tree $T$ corresponds to the total number of mutations captured in $T$.

In Figure 5, we compare the performance of all methods on 6 datasets introduced by Zhou et al. [2024], in terms of the correlation between the terminating state log-probabilities $\log \pi(T)$ associated with the learned policy, and the (uncorrected) return $-\mathcal{E}(T)$, since we should ideally have

$$\log \pi(T) \approx \log \pi^*(T) = -\mathcal{E}(T) - \log Z, \quad (15)$$

based on Theorem 3.1. We observe once again that TB & PCL perform overall similarly to one another, as well as DB & SQL, confirming our observations made above, this time on larger problems where the partition function is intractable. Similar to Section 5.2, we also found that SAC was often less competitive than DB/SQL, except for DS4.

## 6 DISCUSSION

**Stochastic environments.** Similar to [Mohammadpour et al., 2024], we note that our results here are limited to the case where the soft MDP is deterministic, to match the standard assumptions made in the GFlowNet literature. Although some works have attempted to generalize GFlowNets to stochastic environments [Bengio et al., 2023, Pan et al., 2023b], there is no apparent consensus on how to guarantee

the existence of an optimal policy $P_F$ whose terminating state distribution matches (1) when an object may be generated in multiple ways (*i.e.*, $\mathcal{G}$ is a DAG). Jiralerspong et al. [2024] introduced an extension to stochastic environments called *EFlowNet* where an optimal policy is guaranteed to exist, albeit limited to the case where the soft MDP has a tree structure and would therefore bypass the need for any reward correction. The assumption of determinism is generally not limiting though since the structure of the soft MDP is typically designed by an expert based on the problem at hand (depending on the distribution to be approximated).

**Environments with equal number of trajectories.** We saw in Section 2.2 that without correcting the reward function, the optimal policy $\pi^*_{\mathrm{MaxEnt}}$ has a terminating state distribution biased towards states with more complete trajectories leading to them. However, there are some situations where *all* the states have an equal number of trajectories leading to them. This is the case of the discrete factor graphs environments studied in Section 5.1 for example, where all terminating states can be accessed with exactly $n!$ trajectories since each of the $n$ variables can be assigned a value in any order. In this situation, one can apply MaxEnt RL with the original reward in (2), without any correction, and still obtain a terminating state distribution equal to the Gibbs distribution, since this constant can be absorbed into the partition function. Just like the case where $\mathcal{G}$ is a tree, this must be considered as a special case though, and it is generally recommended to always correct the reward with $P_B$.

**Unified parametrization of the policy & state flow.** In all applications of GFlowNets involving the SubTB objective [Madan et al., 2023, Malkin et al., 2022] found in the literature, the forward transition probabilities $P_F^\theta$ and the state flow function $F_\phi$ have always been parametrized with separate models (with possibly a shared backbone) as in Section 3.2. However, thanks to the equivalence with PCL established in Proposition 3.2, it is actually possible

|      | TB     | PCL    | DB         | SQL        | SAC        |
|------|--------|--------|------------|------------|------------|
| DS1  | 0.7797 | 0.7399 | **0.9141** | 0.8695     | 0.6003     |
| DS2  | 0.8550 | 0.8309 | 0.8811     | **0.8922** | 0.7022     |
| DS3  | 0.5833 | 0.6137 | **0.8649** | 0.8474     | 0.6334     |
| DS4  | 0.9178 | 0.9177 | 0.9285     | 0.8965     | **0.9320** |
| DS5  | 0.9688 | 0.9690 | 0.9633     | **0.9712** | 0.9567     |
| DS6  | 0.9526 | 0.9542 | 0.9496     | **0.9615** | 0.8017     |

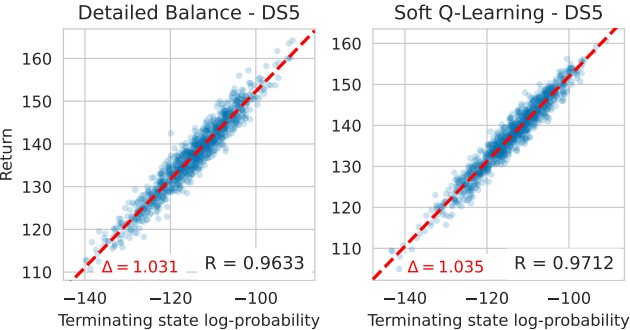

Figure 5: Comparison of MaxEnt RL and GFlowNet algorithms on the phylogenetic tree generation task. (Left) Comparison of the performance in terms of the Pearson correlation coefficient between the terminating state log-probability and the return on 1000 randomly sampled trees. (Center) Correlation between the terminating state log-probability found with DB and the return, each point representing a tree, with a best linear fit line and its slope. (Right) Similarly for SQL. The correlation plots for all methods and all datasets are available in Appendix C.3.

to parametrize both functions using a single "Q-function" thanks to the Unified PCL perspective [Nachum et al., 2017]; see also Appendix B.2 in the context of DB & SQL. A setting with separate networks is closer to the use of a dueling architecture as highlighted by Tiapkin et al. [2024]. In our experiments, we observed that using DB with two separate networks performs overall similarly to SQL with a single Q-network, even though the latter requires fewer parameters to learn. A complete study of the effectiveness of this strategy compared to separate networks is left as future work.

**Continuous control & probabilistic inference.** While this paper focused on discrete distributions, to match the well-studied setting in the GFlowNet literature, the reward correction in Theorem 3.1 may be extended to cases where $\mathcal{X}$, along with the state and action spaces of the soft MDP, are continuous spaces as conjectured by Tiapkin et al. [2024]. We could establish similar connections with MaxEnt RL objectives by leveraging the extensions of GFlowNets to continuous spaces [Li et al., 2023, Lahlou et al., 2023]. Interestingly, continuous GFlowNets have strong connections with diffusion models [Zhang et al., 2022a, Sendera et al., 2024], for which RL has been shown to be an effective training method [Fan and Lee, 2023, Black et al., 2024].

## 7 CONCLUSION & FUTURE WORK

In this work, we showed that many of the well established objectives from the GFlowNet literature [Bengio et al., 2023, Malkin et al., 2022, Pan et al., 2023a] happen to be equivalent to well-known algorithms in MaxEnt RL. Our work is anchored in the recent line of work drawing connections between GFlowNets and MaxEnt RL [Tiapkin et al., 2024, Mohammadpour et al., 2024, Niu et al., 2024], extending the reward correction introduced by Tiapkin et al. [2024] to be applicable at the level of complete trajectories. This

generalization is significant as it allowed us to establish new connections between MaxEnt RL algorithms and GFlowNet objectives, especially in the presence of intermediate rewards in the underlying MDP.

This perspective makes it possible to integrate in a principled way all the tools from Reinforcement Learning for probabilistic inference over large-scale discrete and structured spaces. Future work should be dedicated to further investigating this intersection, borrowing best practices from the RL literature to enable more efficient inference in that setting. In particular, being able to better explore the state space in order to find modes of the Gibbs distribution [Rector-Brooks et al., 2023], for example using targeted exploration strategies developed for RL agents [Bellemare et al., 2016, Pathak et al., 2017], would be essential for large-scale applications such as causal structure learning [Deleu et al., 2022] and molecule generation [Bengio et al., 2021], going beyond the capabilities of MaxEnt RL alone in terms of exploration [Haarnoja et al., 2017].

## ACKNOWLEDGEMENTS

We would like to thank Valentin Thomas, Michal Valko, Pierre Ménard, Daniil Tiapkin, and Sobhan Mohammadpour for helpful discussions and comments about this paper. Doina Precup and Yoshua Bengio are CIFAR Senior Fellows. This research was enabled in part by compute resources and software provided by Mila (mila.quebec).

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

# Discrete Probabilistic Inference as Control in Multi-path Environments (Supplementary Material)

**Tristan Deleu**[1,3]     **Padideh Nouri**[2]     **Nikolay Malkin**[1]     **Doina Precup**[2,4]     **Yoshua Bengio**[1]

Mila – Quebec AI Institute

[1]Université de Montréal   [2]McGill University   [3]Valence Labs   [4]Google DeepMind

## A   REWARD CORRECTION

**Theorem 3.1** (Gen. of Tiapkin et al., 2024; Theorem 1). *Let $P_B(\cdot \mid s')$ be an arbitrary backward transition probability (i.e., a distribution over the parents of $s' \neq s_0$ in $\mathcal{G}$). Let $r(s, s')$ be the reward function of the MDP corrected with $P_B$, satisfying for any trajectory $\tau = (s_0, s_1, \ldots, s_T, s_f)$:*

$$\sum_{t=0}^{T} r(s_t, s_{t+1}) = -\mathcal{E}(s_T) + \alpha \sum_{t=0}^{T-1} \log P_B(s_t \mid s_{t+1}), \tag{8}$$

*where we used the convention $s_{T+1} = s_f$. Then the terminating state distribution associated with the optimal policy $\pi^*_{\mathrm{MaxEnt}}$ solution of (3) satisfies $\pi^*(x) \propto \exp(-\mathcal{E}(x)/\alpha)$.*

*Proof.* Recall from Ziebart [2010], Haarnoja et al. [2017] that the optimal policy maximizing (3) is

$$\pi^*_{\mathrm{MaxEnt}}(s' \mid s) = \exp\left(\frac{1}{\alpha}\left(Q^*_{\mathrm{soft}}(s, s') - V^*_{\mathrm{soft}}(s)\right)\right), \tag{16}$$

where the soft value functions $Q^*_{\mathrm{soft}}$ and $V^*_{\mathrm{soft}}$ satisfy the soft Bellman optimality equations, adapted to our deterministic soft MDP:

$$Q^*_{\mathrm{soft}}(s, s') = r(s, s') + V^*_{\mathrm{soft}}(s') \tag{17}$$

$$V^*_{\mathrm{soft}}(s') = \alpha \log \sum_{s'' \in \mathrm{Ch}(s')} \exp\left(\frac{1}{\alpha} Q^*_{\mathrm{soft}}(s', s'')\right). \tag{18}$$

By definition of the terminating state distribution associated with $\pi^*_{\mathrm{MaxEnt}}$ in (5), for any terminating state $x \in \mathcal{X}$:

$$\pi^*(x) = \sum_{\tau : s_0 \rightsquigarrow x} \prod_{t=0}^{T_\tau} \pi^*_{\mathrm{MaxEnt}}(s_{t+1} \mid s_t) \tag{19}$$

$$= \sum_{\tau : s_0 \rightsquigarrow x} \exp\left[\frac{1}{\alpha} \sum_{t=0}^{T_\tau} \left(Q^*_{\mathrm{soft}}(s_t, s_{t+1}) - V^*_{\mathrm{soft}}(s_t)\right)\right] \tag{20}$$

$$= \sum_{\tau : s_0 \rightsquigarrow x} \exp\left[\frac{1}{\alpha} \sum_{t=0}^{T_\tau} \left(r(s_t, s_{t+1}) + V^*_{\mathrm{soft}}(s_{t+1}) - V^*_{\mathrm{soft}}(s_t)\right)\right] \tag{21}$$

$$= \sum_{\tau : s_0 \rightsquigarrow x} \exp\left[\frac{1}{\alpha}\left(\sum_{t=0}^{T_\tau} r(s_t, s_{t+1}) + \underbrace{V^*_{\mathrm{soft}}(s_f)}_{= 0} - V^*_{\mathrm{soft}}(s_0)\right)\right] \tag{22}$$

$$= \sum_{\tau:s_0 \rightsquigarrow x} \exp \left[ \frac{1}{\alpha} \left( -\mathcal{E}(x) + \alpha \sum_{t=0}^{T_\tau - 1} \log P_B(s_t \mid s_{t+1}) - V_{\text{soft}}^*(s_0) \right) \right] \tag{23}$$

$$= \exp \left[ \frac{1}{\alpha}(-\mathcal{E}(x) - V_{\text{soft}}^*(s_0)) \right] \underbrace{\sum_{\tau:s_0 \rightsquigarrow x} \prod_{t=0}^{T_\tau - 1} P_B(s_t \mid s_{t+1})}_{=1} \tag{24}$$

$$= \frac{\exp(-\mathcal{E}(x)/\alpha)}{\exp(V_{\text{soft}}^*(s_0)/\alpha)} \propto \exp(-\mathcal{E}(x)/\alpha), \tag{25}$$

where we used in (24) the fact that $P_B$ induces a probability distribution over the complete trajectories leading to any terminating state $x$; see for example [Bengio et al., 2023, Lemma 5] for a proof of this result. $\square$

## B  EQUIVALENCES BETWEEN MAXENT RL & GFLOWNET OBJECTIVES

In this section, we detail all of our new results establishing equivalences between MaxEnt RL and GFlowNet objectives, along with their proofs. We summarize the results with links to the propositions and proofs in Table 1. All the objectives considered in this paper take the form of an (expected) least-square $\mathcal{L}(\cdot) = \frac{1}{2}\mathbb{E}_{\pi_b}[\Delta^2(\cdot)]$, where $\Delta(\cdot)$ is a residual term that is algorithm-dependent, and $\pi_b$ is a distribution over appropriate quantities (*i.e.*, trajectories, or transitions); see Section 3.2 for an example with the PCL & SubTB objectives. In this section, we will work exclusively with residuals for simplicity, instead of the objectives themselves. We summarize the different residuals and their correspondences in Figure 6.

Table 1: Summary of the equivalence results and their proofs in Appendix B.

| MaxEnt RL | GFlowNet | Proposition | Proof |
|---|---|---|---|
| PCL [Nachum et al., 2017] | SubTB [Madan et al., 2023] | Proposition 3.2 | App. B.1 |
| SQL [Haarnoja et al., 2017] | DB [Bengio et al., 2023] | Corollary B.1 | App. B.2 |
| SQL$^\star$ [Haarnoja et al., 2017] | FL-DB [Pan et al., 2023a] | Proposition B.3 | App. B.4 |
| $\pi$-SQL$^\star$ (Section 3.3) | M-DB [Deleu et al., 2022] | Proposition 3.3 | App. B.3 |

### B.1  EQUIVALENCE BETWEEN PCL & SUBTB

**Proposition 3.2.** *The Subtrajectory Balance objective (GFlowNet; Madan et al., 2023) is proportional to the Path Consistency Learning objective (MaxEnt RL; Nachum et al., 2017) on the soft MDP with the reward function defined in (9), in the sense that $\mathcal{L}_{\text{PCL}}(\theta, \phi) = \alpha^2 \mathcal{L}_{\text{SubTB}}(\theta, \phi)$, with the following correspondence*

$$\pi_\theta(s' \mid s) = P_F^\theta(s' \mid s) \qquad\qquad V_{\text{soft}}^\phi(s) = \alpha \log F_\phi(s). \tag{11}$$

*Proof.* Let $\tau = (s_m, s_{m+1}, \ldots, s_n)$ be a subtrajectory, where $s_n$ may be the terminal state $s_f$. We first recall the definitions of the Path Consistency Learning (PCL; Nachum et al., 2017) and the Subtrajectory Balance (SubTB; Madan et al., 2023, Malkin et al., 2022) objectives. On the one hand, the PCL objective encourages the consistency between a policy $\pi_\theta$ parametrized by $\theta$ and a value function $V_{\text{soft}}^\phi$ parametrized by $\phi$:

$$\Delta_{\text{PCL}}(\tau; \theta, \phi) = -V_{\text{soft}}^\phi(s_m) + V_{\text{soft}}^\phi(s_n) + \sum_{t=m}^{n-1} \left( r(s_t, s_{t+1}) - \alpha \log \pi_\theta(s_{t+1} \mid s_t) \right). \tag{26}$$

On the other hand, the SubTB objective also encourages some form of consistency, but this time between a policy (forward transition probability) $P_F^\theta$ parametrized by $\theta$ and a flow function $F_\phi$ parametrized by $\phi$. We will give the form of its residual further down, as it depends on the trajectory $\tau$. Finally, recall that the reward function of the soft MDP is defined by (9), in order to satisfy the reward correction necessary for the application of Theorem 3.1, following the same decomposition as in [Tiapkin et al., 2024]

$$r(s_t, s_{t+1}) = \alpha \log P_B(s_t \mid s_{t+1}) \qquad\qquad r(s_T, s_f) = -\mathcal{E}(s_T). \tag{27}$$

Figure 6: Summary of the equivalences between the MaxEnt RL (top, in each box) and GFlowNet (bottom, in each box) objectives, using the classification of Figure 2. All objectives can be written as $\mathcal{L}(\cdot) = \frac{1}{2}\mathbb{E}_{\pi_b}[\Delta^2(\cdot)]$, where $\pi_b$ is a distribution over appropriate quantities (*i.e.*, trajectories, or transitions). The *terminal reward* setting corresponds to $r(s_t, s_{t+1}) = \alpha \log P_B(s_t \mid s_{t+1})$ & $r(s_T, s_f) = -\mathcal{E}(s_T)$ (Section 3.2), whereas the *intermediate rewards* setting corresponds to $r(s_t, s_{t+1}) = -\mathcal{E}(s_t \to s_{t+1}) + \alpha \log P_B(s_t \mid s_{t+1})$ (with $\mathcal{E}(s_t \to s_{t+1}) = \mathcal{E}(s_{t+1}) - \mathcal{E}(s_t)$ if $\mathcal{S} \equiv \mathcal{X}$, Section 3.3) & $r(s_T, s_f) = 0$ (Appendix B.4)

In order to show the equivalence between $\mathcal{L}_{\text{PCL}}$ and $\mathcal{L}_{\text{SubTB}}$, we only need to show equivalence of their corresponding residuals, by replacing the reward by its definition above. We will use the correspondence in (11) between the policy/value function of PCL and the policy/flow function of SubTB. We consider two cases:

- If $s_n \neq s_f$ is not the terminal state, then the residual for SubTB can be written as

$$\Delta_{\text{SubTB}}(\tau; \theta, \phi) = \log \frac{F_\phi(s_n) \prod_{t=m}^{n-1} P_B(s_t \mid s_{t+1})}{F_\phi(s_m) \prod_{t=m}^{n-1} P_F^\theta(s_{t+1} \mid s_t)}, \tag{28}$$

where $P_B$ is a backward transition probability. Although it is in general possible to learn $P_B$ [Malkin et al., 2022], we will consider it fixed here. Substituting (11) into the residual $\Delta_{\text{PCL}}$:

$$\Delta_{\text{PCL}}(\tau; \theta, \phi) = -V_{\text{soft}}^\phi(s_m) + V_{\text{soft}}^\phi(s_n) + \alpha \sum_{t=m}^{n-1} \big( \log P_B(s_t \mid s_{t+1}) - \log \pi_\theta(s_{t+1} \mid s_t) \big)$$

$$= -\alpha \log F_\phi(s_m) + \alpha \log F_\phi(s_n) + \alpha \sum_{t=m}^{n-1} \big( \log P_B(s_t \mid s_{t+1}) - \log P_F^\theta(s_{t+1} \mid s_t) \big)$$

$$= \alpha \log \frac{F_\phi(s_n) \prod_{t=m}^{n-1} P_B(s_t \mid s_{t+1})}{F_\phi(s_m) \prod_{t=m}^{n-1} P_F^\theta(s_{t+1} \mid s_t)} = \alpha \Delta_{\text{SubTB}}(\tau; \theta, \phi). \tag{29}$$

- If $s_n = s_f$, then the residual $\Delta_{\text{SubTB}}$ appearing in the Subtrajectory Balance objective must be written as

$$\Delta_{\text{SubTB}}(\tau; \theta, \phi) = \log \frac{\exp(-\mathcal{E}(s_{n-1})/\alpha) \prod_{t=m}^{n-2} P_B(s_t \mid s_{t+1})}{F_\phi(s_m) \prod_{t=m}^{n-1} P_F^\theta(s_{t+1} \mid s_t)}, \tag{30}$$

since the boundary conditions must also be enforced [Malkin et al., 2022]. Moreover, by definition of the value function, we can also enforce that $V_{\text{soft}}^\phi(s_f) = 0$. Therefore

$$\Delta_{\text{PCL}}(\tau; \theta, \phi) = -V_{\text{soft}}^\phi(s_m) + V_{\text{soft}}^\phi(s_f) - \big( \mathcal{E}(s_{n-1}) + \alpha \log \pi_\theta(s_f \mid s_{n-1}) \big)$$

$$+ \alpha \sum_{t=m}^{n-2} \big( \log P_B(s_t \mid s_{t+1}) - \log \pi_\theta(s_{t+1} \mid s_t) \big) \tag{31}$$

$$= -\alpha \log F_\phi(s_m) - \mathcal{E}(s_{n-1}) - \alpha \log P_F^\theta(s_f \mid s_{n-1})$$

$$+ \alpha \sum_{t=m}^{n-2} \big( \log P_B(s_t \mid s_{t+1}) - \log P_F^\theta(s_{t+1} \mid s_t) \big) \tag{32}$$

$$= \alpha \Delta_{\text{SubTB}}(\tau; \theta, \phi). \tag{33}$$

Note that the Trajectory Balance objective, operating only at the level of complete trajectories [Malkin et al., 2022], corresponds to this case where $s_m = s_0$ is the initial state.

This concludes the proof, showing that $\mathcal{L}_{\text{PCL}}(\theta, \phi) = \alpha^2 \mathcal{L}_{\text{SubTB}}(\theta, \phi)$. $\qquad\square$

## B.2 EQUIVALENCE BETWEEN SQL & DB

The following result establishing the equivalence between the objectives in SQL and DB can be seen as a direct consequence of Proposition 3.2, under the Unified PCL perspective of Nachum et al. [2017]. We state and prove this as a standalone result for completeness.

**Corollary B.1.** *The Detailed Balance objective (GFlowNet; Bengio et al., 2023) is proportional to the Soft Q-Learning objective (MaxEnt RL; Haarnoja et al., 2017) on the soft MDP with the reward function defined in (9), in the sense that* $\mathcal{L}_{\text{SQL}}(\theta) = \alpha^2 \mathcal{L}_{\text{DB}}(\theta)$, *with the following correspondence*

$$F_\theta(s) = \sum_{s'' \in \text{Ch}(s)} \exp\left( \frac{1}{\alpha} Q_{\text{soft}}^\theta(s, s'') \right) \qquad\qquad P_F^\theta(s' \mid s) \propto \exp\left( \frac{1}{\alpha} Q_{\text{soft}}^\theta(s, s') \right). \tag{34}$$

*Proof.* Let $s \to s'$ be a transition in the soft MDP, where $s'$ may be the terminal state. Recall that the residual in the SQL objective depends on a Q-function $Q_{\text{soft}}^\theta$ parametrized by $\theta$:

$$\Delta_{\text{SQL}}(s, s'; \theta) = Q_{\text{soft}}^\theta(s, s') - \left(r(s, s') + V_{\text{soft}}^\theta(s')\right), \tag{35}$$

$$\text{where} \quad V_{\text{soft}}^\theta(s') \triangleq \alpha \log \sum_{s'' \in \text{Ch}(s')} \exp\left(\frac{1}{\alpha} Q_{\text{soft}}^\theta(s', s'')\right). \tag{36}$$

In the case where $s' = s_f$ is the terminal state, then $V_{\text{soft}}^\theta(s') = 0$. On the other hand, the exact form of the residual in the DB objective will be given further down, but always depends on a forward transition probability $P_F^\theta$ and a state flow function $F_\theta$, which we assume are sharing parameters $\theta$. We consider two cases:

- If $s' \neq s_f$ is not the terminal state, then the residual in the DB objective is given by

$$\Delta_{\text{DB}}(s, s'; \theta) = \log \frac{F_\theta(s) P_F^\theta(s' \mid s)}{F_\theta(s') P_B(s \mid s')}. \tag{37}$$

With the definition of the reward function in (9), we know that $r(s, s') = \alpha \log P_B(s \mid s')$. We can therefore show that the residuals of SQL and DB are proportional to one-another:

$$\Delta_{\text{SQL}}(s, s'; \theta) = Q_{\text{soft}}^\theta(s, s') - r(s, s') - \alpha \log \sum_{s'' \in \text{Ch}(s')} \exp\left[\frac{1}{\alpha} Q_{\text{soft}}^\theta(s', s'')\right] \tag{38}$$

$$= Q_{\text{soft}}^\theta(s, s') - \alpha \log P_B(s \mid s') - \alpha \log \sum_{s'' \in \text{Ch}(s')} \exp\left[\frac{1}{\alpha} Q_{\text{soft}}^\theta(s', s'')\right] \tag{39}$$

$$= \alpha \log \left(\exp\left(\frac{1}{\alpha} Q_{\text{soft}}^\theta(s, s')\right)\right) - \alpha \log P_B(s \mid s') - \alpha \log \sum_{s'' \in \text{Ch}(s')} \exp\left[\frac{1}{\alpha} Q_{\text{soft}}^\theta(s', s'')\right]$$

$$- \alpha \log \sum_{s'' \in \text{Ch}(s)} \exp\left[\frac{1}{\alpha} Q_{\text{soft}}^\theta(s, s'')\right] + \alpha \log \sum_{s'' \in \text{Ch}(s)} \exp\left[\frac{1}{\alpha} Q_{\text{soft}}^\theta(s, s'')\right] \tag{40}$$

$$= \alpha \left[\log P_F^\theta(s' \mid s) + \log F_\theta(s) - \log P_B(s \mid s') - \log F_\theta(s')\right] \tag{41}$$

$$= \alpha \Delta_{\text{DB}}(s, s'; \theta), \tag{42}$$

where we used the following correspondence between $Q_{\text{soft}}^\theta$, $P_F^\theta$, and $F_\theta$:

$$F_\theta(s') = \sum_{s \in \text{Pa}(s')} \exp\left(\frac{1}{\alpha} Q_{\text{soft}}^\theta(s, s')\right) \qquad P_F^\theta(s' \mid s) \propto \exp\left(\frac{1}{\alpha} Q_{\text{soft}}^\theta(s, s')\right). \tag{43}$$

- If $s' = s_f$ is the terminal state, then the residual in the DB objective is

$$\Delta_{\text{DB}}(s, s_f; \theta) = \log \frac{F_\theta(s) P_F^\theta(s_f \mid s)}{\exp(-\mathcal{E}(s)/\alpha)}. \tag{44}$$

Again with our definition of the reward function of the soft MDP in (9), we know that the reward of the terminating transition is $r(s, s_f) = -\mathcal{E}(s)$. We can therefore also show the relation between the two residuals in this case:

$$\Delta_{\text{SQL}}(s, s_f; \theta) = Q_{\text{soft}}^\theta(s, s_f) - r(s, s_f) \tag{45}$$

$$= Q_{\text{soft}}^\theta(s, s_f) + \mathcal{E}(s) \tag{46}$$

$$= \alpha \log \left(\exp\left(\frac{1}{\alpha} Q_{\text{soft}}^\theta(s, s_f)\right)\right) + \mathcal{E}(s) \tag{47}$$

$$- \alpha \log \sum_{s'' \in \text{Ch}(s)} \exp\left[\frac{1}{\alpha} Q_{\text{soft}}^\theta(s, s'')\right] + \alpha \log \sum_{s'' \in \text{Ch}(s)} \exp\left[\frac{1}{\alpha} Q_{\text{soft}}^\theta(s, s'')\right]$$

$$= \alpha \left[\log P_F^\theta(s_f \mid s) + \log F_\theta(s) + \frac{\mathcal{E}(s)}{\alpha}\right] \tag{48}$$

$$= \alpha \Delta_{\text{DB}}(s, s_f; \theta), \tag{49}$$

where we used the same correspondence between $Q_{\text{soft}}^\theta$, $P_F^\theta$, and $F_\theta$ as above.

This concludes the proof, showing that $\mathcal{L}_{\text{SQL}}(\theta) = \alpha^2 \mathcal{L}_{\text{DB}}(\theta)$. $\qquad\square$

Note that Tiapkin et al. [2024] established a similar connection between SQL and DB through a dueling architecture perspective [Wang et al., 2016], where the Q-function must be decomposed as (with notation adapted to this paper)

$$Q^\theta(s, s') = V^\theta(s) + A^\theta(s, s') - \log \sum_{s'' \in \text{Ch}(s)} \exp\left(A^\theta(s, s'')\right), \tag{50}$$

where $A^\theta(s, s')$ is an advantage function. The connection they establish is then between $V^\theta$ and the state flow on the one hand, and $A^\theta$ and the policy on the other hand, via

$$\log F_\theta(s) = V^\theta(s) = \log \sum_{s'' \in \text{Ch}(s)} \exp\left(Q^\theta(s, s'')\right) \tag{51}$$

$$\log P_F^\theta(s' \mid s) = A^\theta(s, s') - \log \sum_{s'' \in \text{Ch}(s)} \exp\left(A^\theta(s, s'')\right). \tag{52}$$

This differs from our result in that Corollary B.1 does not explicitly require a separate advantage function. However, we note that both results are effectively equivalent to one another, since (51) directly corresponds to (34) in Corollary B.1 (with $\alpha = 1$), and starting from (52):

$$\log P_F^\theta(s' \mid s) = A^\theta(s, s') - \log \sum_{s'' \in \text{Ch}(s)} \exp\left(A^\theta(s, s'')\right) \tag{53}$$

$$= \left[Q^\theta(s, s') - V^\theta(s) + \log \sum_{s'' \in \text{Ch}(s)} \exp\left(A^\theta(s, s'')\right)\right] - \log \sum_{s'' \in \text{Ch}(s)} \exp\left(A^\theta(s, s'')\right)$$

$$= Q^\theta(s, s') - V^\theta(s) = Q^\theta(s, s') - \log \sum_{s'' \in \text{Ch}(s)} \exp\left(Q^\theta(s, s'')\right), \tag{54}$$

which also corresponds to (34) in Corollary B.1. The connection through a dueling architecture is closer to DB when the policy and the state flow network are parametrized by two separate networks.

## B.3   EQUIVALENCE BETWEEN $\pi$-SQL AND MODIFIED DB

**Proposition B.2.** *Assume that all the states of the soft MDP are terminating* (i.e., *connected to the terminal state $s_f$), and such that for all $s \in \mathcal{S}$, the reward function satisfies $r(s, s_f) = 0$. Then the objective of Soft Q-Learning [Haarnoja et al., 2017] can be written as a function of a policy $\pi_\theta$ parametrized by $\theta$. This objective is given by $\mathcal{L}_{\pi\text{-SQL}}(\theta) = \frac{1}{2}\mathbb{E}_{\pi_b}[\Delta^2_{\pi\text{-SQL}}(s, s'; \theta)]$, where $\pi_b$ is an arbitrary policy over transitions $s \to s'$ such that $s' \neq s_f$, and*

$$\Delta_{\pi\text{-SQL}}(s, s'; \theta) = \alpha\left[\log \pi_\theta(s' \mid s) - \log \pi_\theta(s_f \mid s) + \log \pi_\theta(s_f \mid s')\right] - r(s, s'). \tag{55}$$

*Proof.* Recall that the objective of Soft Q-Learning can be written in terms of a Q-function $Q^\theta_{\text{soft}}$ parametrized by $\theta$ as $\mathcal{L}_{\text{SQL}}(\theta) = \frac{1}{2}\mathbb{E}_{\pi_b}[\Delta^2_{\text{SQL}}(s, s'; \theta)]$, with

$$\Delta_{\text{SQL}}(s, s'; \theta) = Q^\theta_{\text{soft}}(s, s') - \left(r(s, s') + V^\theta_{\text{soft}}(s')\right), \tag{56}$$

$$\text{where} \quad V^\theta_{\text{soft}}(s') \triangleq \alpha \log \sum_{s'' \in \text{Ch}(s')} \exp\left[\frac{1}{\alpha}Q^\theta_{\text{soft}}(s', s'')\right]. \tag{57}$$

Since we assume that $r(s, s_f) = 0$, we can enforce the fact that $Q^\theta_{\text{soft}}(s, s_f) = 0$ in our parametrization of the Q-function. If we define a policy $\pi_\theta$ as

$$\pi_\theta(s' \mid s) \triangleq \exp\left[\frac{1}{\alpha}\left(Q^\theta_{\text{soft}}(s, s') - V^\theta_{\text{soft}}(s)\right)\right], \tag{58}$$

then we have in particular $\pi_\theta(s_f \mid s) = \exp(-V^\theta_{\text{soft}}(s)/\alpha)$, based on our observation above. Moreover, we can write the different in value functions appearing in (56) as

$$Q^\theta_{\text{soft}}(s, s') - V^\theta_{\text{soft}}(s') = Q^\theta_{\text{soft}}(s, s') - V^\theta_{\text{soft}}(s) + V^\theta_{\text{soft}}(s) - V^\theta_{\text{soft}}(s') \tag{59}$$

$$= \alpha\left[\log \pi_\theta(s' \mid s) - \log \pi_\theta(s_f \mid s) + \log \pi_\theta(s_f \mid s')\right] \tag{60}$$

which concludes the proof. $\qquad\square$

**Proposition 3.3.** *Suppose that all the states of the soft MDP are terminating $\mathcal{S} \equiv \mathcal{X}$. The Modified Detailed Balance objective (GFlowNet; Deleu et al., 2022) is proportional to the Soft Q-Learning objective with a policy parametrization (MaxEnt RL; $\pi$-SQL) on the soft MDP with the reward function defined in (12), in the sense that $\mathcal{L}_{\pi\text{-SQL}}(\theta) = \alpha^2 \mathcal{L}_{\text{M-DB}}(\theta)$, with $\pi_\theta(s' \mid s) = P_F^\theta(s' \mid s)$.*

*Proof.* Let $s \to s'$ be a transition in the soft MDP. In order to show the equivalence between $\mathcal{L}_{\pi\text{-SQL}}$ and $\mathcal{L}_{\text{M-DB}}$, it is sufficient to show the equivalence between their corresponding residuals (13) and (14). Recall that the reward function in (12) is defined by

$$r(s_t, s_{t+1}) = \mathcal{E}(s_t) - \mathcal{E}(s_{t+1}) + \alpha \log P_B(s_t \mid s_{t+1}) \qquad\qquad r(s_T, s_f) = 0. \qquad (61)$$

Replacing the reward in the residual $\Delta_{\pi\text{-SQL}}$, we get

$$
\begin{aligned}
\Delta_{\pi\text{-SQL}}(s, s'; \theta) &= \alpha\big[ \log \pi_\theta(s' \mid s) - \log \pi_\theta(s_f \mid s) + \log \pi_\theta(s_f \mid s') \big] - r(s, s') \\
&= \alpha\big[ \log \pi_\theta(s' \mid s) - \log \pi_\theta(s_f \mid s) + \log \pi_\theta(s_f \mid s') \big] - \big[ \mathcal{E}(s) - \mathcal{E}(s') + \alpha \log P_B(s \mid s') \big] \\
&= \alpha\left[ -\frac{\mathcal{E}(s)}{\alpha} + \log \pi_\theta(s' \mid s) + \log \pi_\theta(s_f \mid s') + \frac{\mathcal{E}(s')}{\alpha} - \log P_B(s \mid s') - \log \pi_\theta(s_f \mid s) \right] \\
&= \alpha\left[ -\frac{\mathcal{E}(s)}{\alpha} + \log P_F^\theta(s' \mid s) + \log P_F^\theta(s_f \mid s') + \frac{\mathcal{E}(s')}{\alpha} - \log P_B(s \mid s') - \log P_F^\theta(s_f \mid s) \right] \\
&= -\alpha \log \frac{\exp(-\mathcal{E}(s')/\alpha) P_B(s \mid s') P_F^\theta(s_f \mid s)}{\exp(-\mathcal{E}(s)/\alpha) P_F^\theta(s' \mid s) P_F^\theta(s_f \mid s')} = -\alpha \Delta_{\text{M-DB}}(s, s'; \theta)
\end{aligned}
$$

This conclude the proof, showing that $\mathcal{L}_{\pi\text{-SQL}}(\theta) = \alpha^2 \mathcal{L}_{\text{M-DB}}(\theta)$. $\qquad\square$

## B.4 EQUIVALENCE BETWEEN SQL AND FORWARD-LOOKING DB

We will now generalize the result of Proposition 3.3 to the case where $\mathcal{X} \not\equiv \mathcal{S}$, but where intermediate rewards are still available along the trajectory. We will assume that for any complete trajectory $\tau = (s_0, s_1, \ldots, s_T, s_f)$, the energy function at $s_T$ can be decomposed into a sum of intermediate rewards [Pan et al., 2023a]

$$\mathcal{E}(s_T) = \sum_{t=0}^{T-1} \mathcal{E}(s_t \to s_{t+1}), \qquad\qquad (62)$$

where we overload the notation $\mathcal{E}$ for simplicity. In that case, we can define the corrected reward as follows in order to satisfy the conditions of Theorem 3.1

$$r(s_t, s_{t+1}) = -\mathcal{E}(s_t \to s_{t+1}) + \alpha \log P_B(s_t \mid s_{t+1}) \qquad\qquad r(s_T, s_f) = 0. \qquad (63)$$

This type of reward shaping is similar to the one introduced in Section 3.3. The Forward-Looking Detailed Balance loss (FL-DB; Pan et al., 2023a) is defined similarly to DB, with the exception that the flow function corresponds to the unknown offset relative to $\mathcal{E}(s_t \to s_{t+1})$, which is known and therefore does not need to be learned. For some transition $s \to s'$ such that $s' \neq s_f$, the corresponding residual can be written as

$$\Delta_{\text{FL-DB}}(s, s'; \theta) = \log \frac{\tilde{F}_\theta(s') P_B(s \mid s')}{\tilde{F}_\theta(s) P_F^\theta(s' \mid s)} - \frac{\mathcal{E}(s \to s')}{\alpha}, \qquad\qquad (64)$$

where $P_F^\theta$ is the policy (forward transition probability), and $\tilde{F}_\theta$ is an offset state-flow function, parametrized by $\theta$. Note that with FL-DB, there is no longer an explicit residual for the boundary condition, unlike in DB, since this is captured through (64) already. The following proposition establishes an equivalence between SQL and FL-DB, similar to Corollary B.1 & Proposition 3.3.

**Proposition B.3.** *The Forward-Looking Detailed Balance objective (GFlowNet; Pan et al., 2023a) is proportional to the Soft Q-Learning objective (MaxEnt RL; Haarnoja et al., 2017) on the soft MDP with the reward function defined in (63), in the sense that $\mathcal{L}_{\text{SQL}}(\theta) = \alpha^2 \mathcal{L}_{\text{FL-DB}}(\theta)$, with the following correspondence*

$$\tilde{F}_\theta(s) = \sum_{s'' \in \text{Ch}(s)} \exp\left( \frac{1}{\alpha} Q_{\text{soft}}^\theta(s, s'') \right) \qquad\qquad P_F^\theta(s' \mid s) \propto \exp\left( \frac{1}{\alpha} Q_{\text{soft}}^\theta(s, s') \right). \qquad (65)$$

*Proof.* The proof is similar to the one in Appendix B.2. Let $s \to s'$ be a transition in the soft MDP, where $s' \neq s_f$. Recall that the residual in the SQL objective is

$$\Delta_{\text{SQL}}(s, s'; \theta) = Q^\theta_{\text{soft}}(s, s') - \left( r(s, s') + V^\theta_{\text{soft}}(s') \right), \tag{66}$$

$$\text{where} \quad V^\theta_{\text{soft}}(s') \triangleq \alpha \log \sum_{s'' \in \text{Ch}(s')} \exp\left( \frac{1}{\alpha} Q^\theta_{\text{soft}}(s', s'') \right). \tag{67}$$

With our choice of reward function in (63), we know that $r(s, s') = -\mathcal{E}(s \to s') + \alpha \log P_B(s \mid s')$. We can therefore show that the residuals of SQL and FL-DB are proportional to one-another:

$$\Delta_{\text{SQL}}(s, s'; \theta) = Q^\theta_{\text{soft}}(s, s') - \left( r(s, s') + V^\theta_{\text{soft}}(s') \right) \tag{68}$$

$$= Q^\theta_{\text{soft}}(s, s') + \mathcal{E}(s \to s') - \alpha \log P_B(s \mid s') - \alpha \log \sum_{s'' \in \text{Ch}(s')} \exp\left( \frac{1}{\alpha} Q^\theta_{\text{soft}}(s', s'') \right) \tag{69}$$

$$= Q^\theta_{\text{soft}}(s, s') - \alpha \log \sum_{s'' \in \text{Ch}(s)} \exp\left( \frac{1}{\alpha} Q^\theta_{\text{soft}}(s, s'') \right) + \mathcal{E}(s \to s') - \alpha \log P_B(s \mid s')$$

$$\quad + \alpha \log \sum_{s'' \in \text{Ch}(s)} \exp\left( \frac{1}{\alpha} Q^\theta_{\text{soft}}(s, s'') \right) - \alpha \log \sum_{s'' \in \text{Ch}(s')} \exp\left( \frac{1}{\alpha} Q^\theta_{\text{soft}}(s', s'') \right) \tag{70}$$

$$= \alpha \left[ \log P^\theta_F(s' \mid s) - \log P_B(s \mid s') + \log \tilde{F}_\theta(s) - \log \tilde{F}_\theta(s') + \frac{\mathcal{E}(s \to s')}{\alpha} \right] \tag{71}$$

$$= -\alpha \Delta_{\text{FL-DB}}(s, s'; \theta). \tag{72}$$

Where we used the correspondence between $\tilde{F}_\theta$, $P^\theta_F$, and $Q^\theta_{\text{soft}}$ from (65). This concludes the proof, showing that $\mathcal{L}_{\text{SQL}}(\theta) = \alpha^2 \mathcal{L}_{\text{FL-DB}}(\theta)$. □

Interestingly, the correspondence in (65) between state flows and policy on the one hand (GFlowNet) and the Q-function on the other hand (MaxEnt RL) is exactly the same as the one in Corollary B.1.

## C   EXPERIMENTAL DETAILS

For all algorithms and all environments (unless stated otherwise), we kept the same exploration schedule, frequency of update of the target network (if applicable), and replay buffer, in order to avoid attributing favorable performance to any of those components. Exploration was done using a naive $\varepsilon$-sampling scheme, where actions were sampled from the current policy with probability $1 - \varepsilon$, and uniformly at random with probability $\varepsilon$. All algorithms were trained over 100k iterations, and $\varepsilon$ was decreasing over the first 50k from $\varepsilon = 1$ to $\varepsilon = 0.1$. All algorithms use a target network, except TB/PCL, and the target network was updated every 1000 iterations. We used a simple circular buffer with 100k capacity for all algorithms (TB/PCL using buffer of trajectories, as opposed to a buffer of transitions). Hyperparameter search was conducted for all environments over the learning rate alone of all the networks, using a simple grid search.

### C.1   PROBABILISTIC INFERENCE OVER DISCRETE FACTOR GRAPHS

Given a factor graph with a fixed structure, over $d$ random variables $(V_1, \ldots, V_d)$, the objective is to sample a complete assignment $\boldsymbol{v}$ of these variables from the Gibbs distribution $P(\boldsymbol{v}) \propto \exp(-\mathcal{E}(\boldsymbol{v}))$, where the energy function is defined by

$$\mathcal{E}(v_1, \ldots, v_d) = -\sum_{m=1}^{M} \psi_m(\boldsymbol{v}_{[m]}), \tag{73}$$

where $\psi_m$ is the $m$th factor in the factor graph, and $\boldsymbol{v}_{[m]}$ represents the values of the variables that are part of this factor. The factors $\psi_m$ are fixed, and randomly generated using the same process as Buesing et al. [2020]. Each variable $V_i$ is assumed to be discrete and can take one of $K$ possible values. Overall, this means that the number of elements in the sample space is $K^d$.

A state of the soft MDP is a (possibly partial) assignment of the values, *e.g.*, $(0, \cdot, 1, 0, \cdot, \cdot)$, where $\cdot$ represents a variable which has not been assigned a value yet. The initial state $s_0 = (\cdot, \cdot, \ldots, \cdot)$ is the state where no variable has an assigned value. An action consists in picking one variable that has not value (with a $\cdot$), and assigning it one of $K$ values. The process terminates when all the variables have been assigned a value, meaning that all the complete trajectories have length $d$. This differs from the MDP of Buesing et al. [2020], since they were assigning the values of the variables in a fixed order determined ahead of time, making the MDP having a tree structure.

Buesing et al. [2020] defined an intermediate reward function corresponding to a decomposition of the energy (73) as $\mathcal{E}(\boldsymbol{v}) = \sum_{t=0}^{d-1} \mathcal{E}(s_t \to s_{t+1})$, where $\boldsymbol{v}$ is the terminating state of a complete trajectory $(s_0, s_1, \ldots, s_d, s_f)$ (*i.e.*, $s_d = \boldsymbol{v}$), and with the partial energies defined as

$$\mathcal{E}(s_t \to s_{t+1}) = -\sum_{m=1}^{M} \psi_m(\boldsymbol{v}_{[m]}) \mathbb{1}(i \in [m] \ \& \ \boldsymbol{v}_{[m]} \subseteq s_{t+1}), \tag{74}$$

if the transition $s_t \to s_{t+1}$ corresponds to assigning the value of a particular variable $V_i$. In other words, the partial energy corresponds to computing all the factors as soon as all the necessary information is available (*i.e.*, all the values of the input variables of the factors have been assigned, and $V_i$ that was just assigned a value is one of the input variables of the factors).

## C.2 STRUCTURE LEARNING OF BAYESIAN NETWORKS

Given $d$ continuous random variables $(X_1, \ldots, X_d)$, a DAG $G$ and parameters $\theta$, a Bayesian Network represents the conditional independences in the joint distribution based on the structure of $G$

$$P(X_1, \ldots, X_d; \theta) = \prod_{j=1}^{d} P\big(X_j \mid \mathrm{Pa}_G(X_j); \theta_k\big), \tag{75}$$

where $\mathrm{Pa}_G(X_j)$ represents the parent variables of $X_j$ in $G$. We assume that all conditional distributions are linear-Gaussian. The objective of Bayesian structure learning is to approximate the posterior distribution over DAGs: $P(G \mid \mathcal{D}) \propto P(\mathcal{D} \mid G)P(G)$, where $P(\mathcal{D} \mid G)$ is the marginal likelihood and $P(G)$ is a prior over graph, assumed to be uniform here. Our experiments vary in the way the marginal likelihood is computed, either based on the BGe score [Geiger and Heckerman, 1994], or the linear Gaussian score [Nishikawa-Toomey et al., 2023]. We followed the experimental setup of Deleu et al. [2022], where data is generated from a randomly generated ground truth Bayesian network $G^*$, sampled using an Erdös-Rényi scheme with on average 1 edge per node. We generated 100 observations from this Bayesian Network using ancestral sampling. We repeated this process for 20 different random seeds.

A state of the soft MDP corresponds to a DAG $G$ over $d$ nodes, and the initial state is the empty graph over $d$ nodes. An action consists in adding a directed edge between two nodes, such that it is not already present in the graph, and it doesn't introduce a cycle, garanteeing that all the states of the MDP are valid acyclic graphs; there is a special action indicating whether we want to terminate and transition to $s_f$. Since all the states are valid DAGs, this means that all the states of the MDP are terminating, and we can use $r(G_t, G_{t+1}) = \mathcal{E}(G_t) - \mathcal{E}(G_{t+1})$ as the intermediate reward, with the appropriate energy function $\mathcal{E}(G) = \log P(G_0, \mathcal{D}) - \log P(G, \mathcal{D})$. Deleu et al. [2022] showed that the difference in energies can be efficiently computed using the delta-score [Friedman and Koller, 2003].

## C.3 PHYLOGENETIC TREE GENERATION

We consider the environment introduced by Zhou et al. [2024], where phylogenertic trees used for the analysis of the evolution of a group of $d$ species are generated, according to a parsimonious criterion. Indeed, trees encoding few mutations are favored as they are more likely to represent realistic relationships. Given a tree $T$, whose nodes are the species of interest, the target distribution is given by

$$P(T) \propto \exp(-M(T \mid \boldsymbol{Y})/C), \tag{76}$$

where $C = 4$ is a fixed constant, and $M(T \mid \boldsymbol{Y})$ is the total number of mutations (also known as the *parsimony score*), based on the biological sequences $\boldsymbol{Y}$ associated with each species; note that for convenience, we treat the energy function as being $\mathcal{E}(T) = M(T \mid \boldsymbol{Y})/C$ (with $\alpha = 1$). We used 6 out of the 8 datasets considered by Zhou et al. [2024]; the statistics of the datasets are recalled in Table 2 for completeness.

Table 2: Statistics of the datasets used in the phylogenetic tree generation task. "Length" represents the length of the biological sequence (*e.g.*, the DNA sequence) of each species. See [Zhou et al., 2024] for details and references about these datasets. The number of species represents the number of nodes in the tree, and is a measure of complexity of the task.

| Dataset | # Species ($d$) | Length |
|---------|-----------------|--------|
| DS1 | 27 | 1949 |
| DS2 | 29 | 2520 |
| DS3 | 36 | 1812 |
| DS4 | 41 | 1137 |
| DS5 | 50 | 378 |
| DS6 | 50 | 1133 |

A state of the soft MDP is a collection of trees over a partition of all the species (the leaves of the trees are species), where the initial state corresponds to $d$ trees with a single node (leaf), one for each species in the group. An action consists in picking two trees, and merging them by adding a root. The process terminates when there is only one tree left in this collection, meaning that all complete trajectories have the same length $d - 1$. The size of the sample space is $(2d - 3)!!$ (for $d \geq 2$).

To decompose the energy function into $\mathcal{E}(T) = \sum_{t=0}^{d-2} \mathcal{E}(s_t \rightarrow s_{t+1})$, where $T$ is the terminating state of a complete trajectory $(s_0, s_1, \ldots, s_{d-1}, s_f)$, we can use the observation made by Zhou et al. [2024] that the total number of mutations $M(T \mid \boldsymbol{Y})$ can be decomposed as the sum of (1) the number of mutations are the root of the tree, and (2) the total number of mutations in the left and right subtrees. Therefore, we can use the number of mutations at the new root of the tree constructed during the transition $s_t \rightarrow s_{t+1}$ as the intermediate energy $\mathcal{E}(s_t \rightarrow s_{t+1})$ (appropriately rescaled by $C$), which can be computed using the Fitch algorithm [Zhou et al., 2024].

In addition to Figure 5, we also provide a complete view of the correlation between the terminating state log-probabilities and the returns for all algorithms and all datasets in Figures 7 & 8. Each point corresponds to a tree sampled using the terminating state distribution found by the corresponding algorithm.

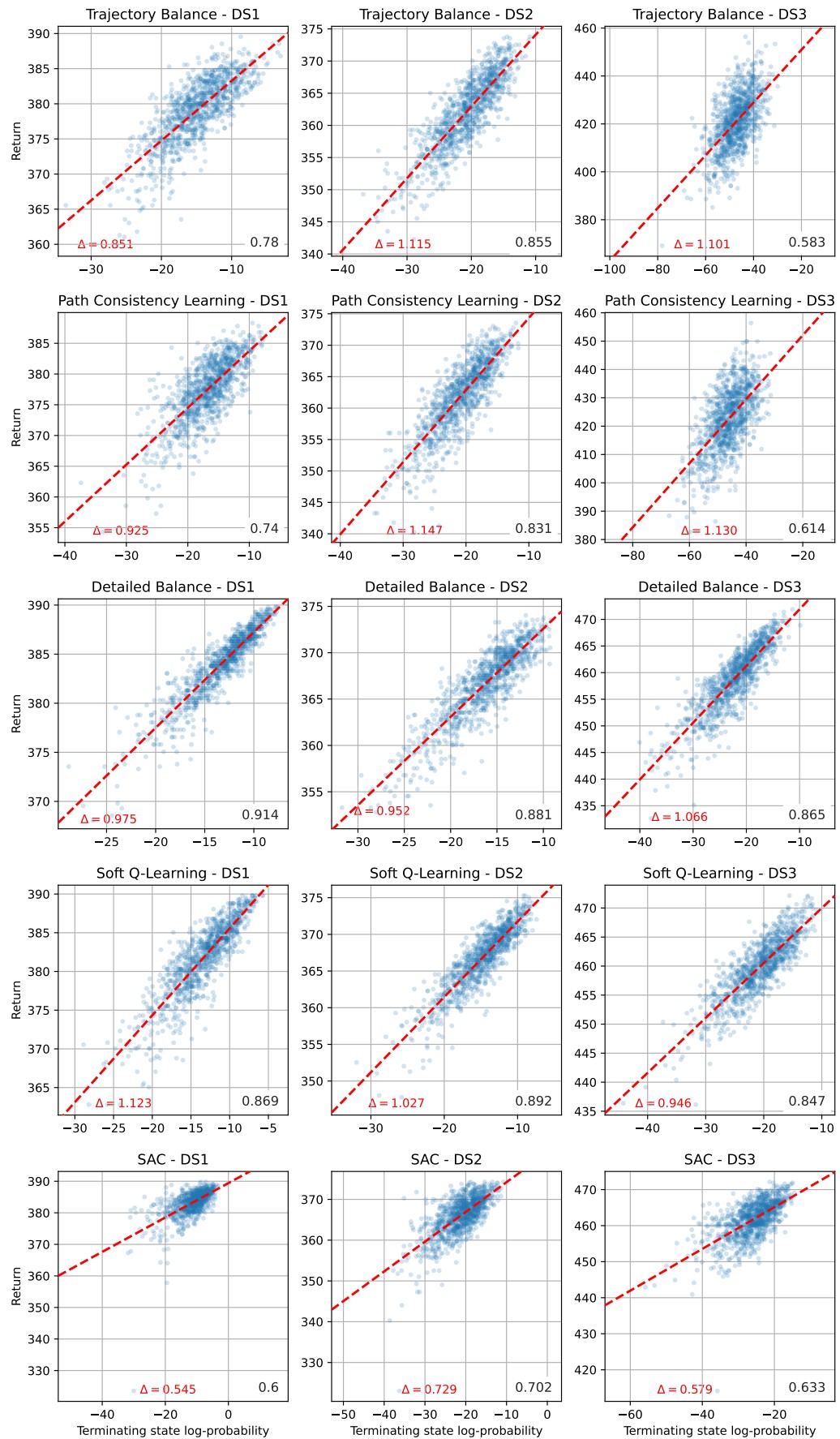

Figure 7: Correlation plots for all algorithms. Rows from top to bottom: Trajectory Balance (TB), Path Consistency Learning (PCL), Detailed Balance (DB), Soft Q-Learning (SQL) and SAC. Columns showing DS1 to DS3 from left to right.

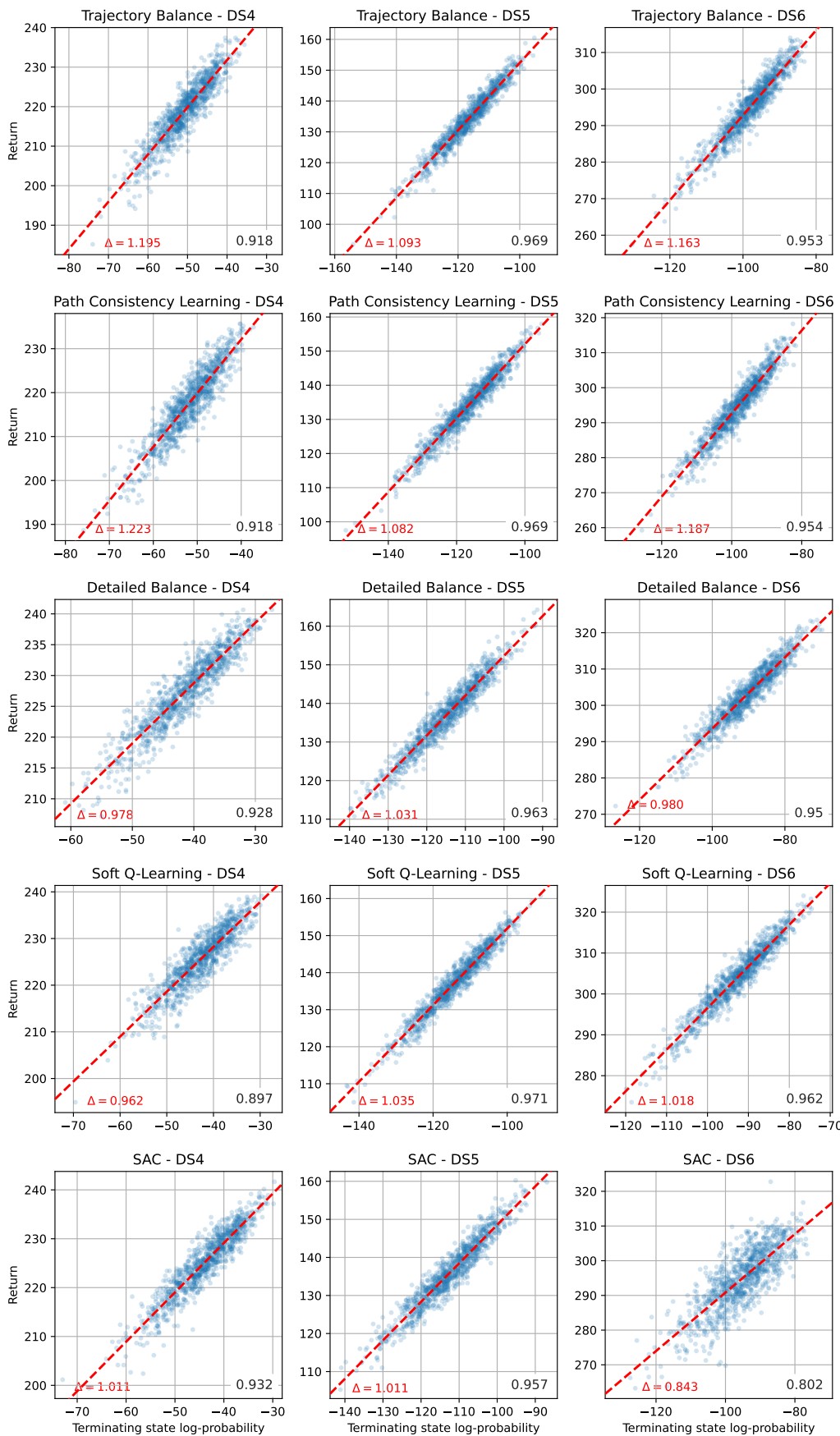

Figure 8: Similar plots as Figure 7 here columns presenting DS4, DS5 and DS6 from left to right.