# OpenReview forum: "Discrete Probabilistic Inference as Control in Multi-path Environments"
_auai.org/UAI/2024/Conference — UAI 2024 poster_

### Official Review · Reviewer_gXAk · 2024-03-22

**Q2-1 Originality-Novelty:** 2
**Q2-2 Correctness-Technical Quality:** 3
**Q2-5 Clarity Of Writing:** 3

**Q1 Summary And Contributions:**

The paper continues a line of work that explores the relationship between two distinct approaches to approximate discrete probabilistic inference via sequential decision making: maximum entropy Reinforcement Learning (MaxEnt RL) and Generative Flow Networks (GFlowNets). First it generalises recent work that links these two methods via reward correction. It then points out that some of the prior work on each model coincides. Finally, some experiments are performed to investigate the performance of various methods for MaxEnt RL and GFlowNets on some benchmark problems.

**Q2-3 Extent To Which Claims Are Supported By Evidence:**

3: Good: the main claims are supported by convincing evidence (in the form of adequate experimental evaluation, proofs, (pseudo-)code, references, assumptions).

**Q2-4 Reproducibility:**

4: Excellent: key resources (e.g. proofs, code, data) are available and key details (e.g. proof sketches, experimental setup) are comprehensively described for competent researchers to confidently and easily reproduce the main results.

**Q3 Main Strengths:**

+ Competent technical contribution on a topic seemingly generating some interest currently

+ Good discussion of related work and the technical similarities/differences

+ Mixture of theoretical and empirical results

+ Comprehensive details provided of proofs and empirical setup

**Q4 Main Weakness:**

- Interesting work but the technical contribution here seems very modest in the light of existing work on exactly the same topic

- It's not clear to me what the overall message of the experimental results is here, nor how it relates to the theoretical parts of the paper

**Q5 Detailed Comments To The Authors:**

Questions:

(1) In Sec 3.3, you consider the case where all states of the soft MDP are terminating, but no motivation is given. Why is this class particularly useful for inference?

(2) How does the theoretical part of this paper (Sec 3) actually enable/link to the experimental part? You seem to just "validate" the techniques already known/proven to be equivalent.

(3) Section 5 presents empirical results for three different types of inference benchmarks, which is a positive. But is there anything to say about the type/structure of these different problems affects the results/performance?

Small fixes:

Sec 2.1: You should introduce the term "soft MDP" here

Sec 2.1: It would help the reader, after equation (3) to mention the intuition behind adding the entropy to states

**Q9 Complying With Reviewing Instructions:**

Yes

---

> ### Author Rebuttal · Authors · 2024-04-01
>
> We would like to thank the reviewer for their review and their questions.
>
> > *Interesting work but the technical contribution here seems very modest in the light of existing work on exactly the same topic*
>
> This paper is the first work to comprehensively study the connections between GFlowNet objectives and MaxEnt RL algorithms, when they have been only alluded to without proof in past work (e.g., [Bengio et al., 2021], [Malkin et al., 2022], [Hu et al., 2024]). This goes further than the recent work of [Tiapkin et al., 2024], which only focuses on the DB/SQL connection in a sparse reward setting, and [Mohammadpour et al., 2024], which connects PCL only with TB — not the more general SubTB — and only in the case of the specific backward policy studied in that paper.
>
> Our generalization of the result of [Tiapkin et al., 2024] is also significant since it allows us to consider cases where intermediate reward is available, which was not possible with prior results. This allowed us to establish for the first time the equivalences $\pi$-SQL/modified DB and SQL/forward-looking DB.
>
> > *It's not clear to me what the overall message of the experimental results is here, nor how it relates to the theoretical parts of the paper*
>
> The goal of these experiments is to empirically support the equivalences we establish in the paper, showing that TB & PCL, and DB & SQL perform identically. This is clearer in Sec. 5.1 & 5.2, where we can compare the JSD with the target distribution, showing overlapping curves. Sec. 5.3, shows that this also holds on large scale problems, where the exact comparison with the target distribution is not feasible anymore.
>
> > *(1) In Sec 3.3, you consider the case where all states of the soft MDP are terminating, but no motivation is given. Why is this class particularly useful for inference?*
>
> The case where all the states are terminating (i.e., valid samples of the target distribution) is often considered in the MCMC literature, where moves are from one sample to another. In fact, the difference in energies from (13) is a quantity that appears in the acceptance probability of Metropolis-Hastings. One difference between MCMC and our setting though is that we consider the MDP to be a DAG, and therefore there is no irreducibility (which is a common assumption in MCMC). The Bayesian structure learning example (Sec 5.2) is an example where all the states are terminating.
>
> > *(2) How does the theoretical part of this paper (Sec 3) actually enable/link to the experimental part? You seem to just "validate" the techniques already known/proven to be equivalent.*
>
> The primary objective is to show that these equivalences are not just theoretical curiosities, but they have strong implications empirically as well. Furthermore, in addition to the algorithms we proved to be equivalent to one another, we also tested SAC empirically, an algorithm that has no known GFlowNet counterpart, on these probabilistic inference problems. Interestingly, although SAC is known to be a strong method in RL, we found that it was unstable empirically on those problems, hopefully inviting the community to further investigate stable variants of SAC for probabilistic inference.
>
> > *(3) Section 5 presents empirical results for three different types of inference benchmarks, which is a positive. But is there anything to say about the type/structure of these different problems affects the results/performance?*
>
> All these problems are chosen intentionally because they allow for intermediate rewards, which is a setting that is not studied in prior connections between GFlowNets & MaxEnt RL. Although TB/PCL (on complete trajectories) would not benefit from the intermediate reward signal, SubTB & DB/SQL would.
>
> > *Small fixes*
>
> Thank you for your suggestions. We agree with you and will implement both of them in the camera ready version of the paper.

---

### Official Review · Reviewer_TPBh · 2024-03-22

**Q2-1 Originality-Novelty:** 3
**Q2-2 Correctness-Technical Quality:** 3
**Q2-5 Clarity Of Writing:** 3

**Q1 Summary And Contributions:**

The paper considers the problem of sampling from a discrete and structured probability distribution as a sequential decision making problem. The latter is assumed to be modelled as an MDP. Over the past few years, two main approaches have emerged in the literature: maximum entropy reinforcement learning (maxed RL) and generative flow networks (gflownets). While both methods learn stochastic policies that are subsequently used for sampling, there are subtle connections between the methods. The paper looks closely at both methods and establishes theoretically the equivalence between some previously proposed GFlowNets objectives and MaxEnt RL algorithms with a corrected reward. These observations are subsequently validated in experiments. The results show clearly that the equivalent methods perform similarly in practice.

**Q2-3 Extent To Which Claims Are Supported By Evidence:**

4: Excellent: all claims are supported by very convincing evidence (in the form of comprehensive experimental evaluation, rigorous mathematical proofs, detailed (pseudo-)code, precise references, well-motivated and realistic assumptions) and the authors deliver what they promise.

**Q2-4 Reproducibility:**

3: Good: key resources (e.g. proofs, code, data) are available and key details (e.g. proofs, experimental setup) are sufficiently well-described for competent researchers to confidently reproduce the main results.

**Q3 Main Strengths:**

The paper considers an important problem in machine learning, namely probabilistic inference as a decision making problem, with potentially many real-world applications. The main contribution consists in establishing a deep connection (equivalence) between the two most popular approaches in the literature by generalising some previous results.

The paper is organised pretty well, enough background is provided and the contribution is presented in a relatively clear manner. Therefore the overall quality of the presentation is fairly good.

The empirical evaluation is sound and the results are presented in a relatively clear manner. Therefore it is easy to grasp the trend of the results and validate the equivalence between the two methods that was established in the main section of the paper.

**Q4 Main Weakness:**

I wasn't able to find any major weakness in the paper. Perhaps, one suggestion is to include some motivating examples for the reader that is not very familiar with GFlowNets and MaxEnt RL approaches.

**Q5 Detailed Comments To The Authors:**

Please see the previous sections for comments.

**Q9 Complying With Reviewing Instructions:**

Yes

---

> ### Author Rebuttal · Authors · 2024-04-01
>
> We would like to thank the reviewer for their review and their positive feedback regarding the clarity of our paper.
>
> > *Perhaps, one suggestion is to include some motivating examples for the reader that is not very familiar with GFlowNets and MaxEnt RL approaches.*
>
> Thank you for your suggestion. With the extra space permitted in the camera ready version, we will consider expanding the discussion in Figure 1 to motivate MaxEnt RL and GFlowNets earlier in the paper. We will also mention in the introduction other settings where both methods have been used. This improved presentation will highlight the way in which our work bridges the typical motivation and setting of MaxEnt RL — control with provable robustness, see, e.g.. [Eysenbach & Levine, 2022] — with the typical motivation and setting of GFlowNets — asymptotically correct sampling of Bayesian posteriors and diverse generation of structured objects (currently discussed in Sec. 4 and App. A.1).

---

### Official Review · Reviewer_vfpH · 2024-03-23

**Q2-1 Originality-Novelty:** 2
**Q2-2 Correctness-Technical Quality:** 3
**Q2-5 Clarity Of Writing:** 2

**Q1 Summary And Contributions:**

The paper establishes theoretical equivalences between GFlownets and maxentropy reinforcement learning. Here the paper generalizes findings found in prior work. Experimentally, the paper then confirms these findings by showing similar performance between GFlownets and equivalent MaEntropy settings.

**Q2-3 Extent To Which Claims Are Supported By Evidence:**

3: Good: the main claims are supported by convincing evidence (in the form of adequate experimental evaluation, proofs, (pseudo-)code, references, assumptions).

**Q2-4 Reproducibility:**

3: Good: key resources (e.g. proofs, code, data) are available and key details (e.g. proofs, experimental setup) are sufficiently well-described for competent researchers to confidently reproduce the main results.

**Q3 Main Strengths:**

The paper establishes clear theoretical connections between GFlownets and MaxEnt RL and these are then confirmed experimentally.

**Q4 Main Weakness:**

The story of the paper seems to be a bit of. The title and abstract do not seem to really match the content of the paper.

**Q5 Detailed Comments To The Authors:**

Not a lot of comments as this is really not my field of expertise.
Nevertheless, could the authors comment on more standard inference techniques for discrete random variables. Specifically, are there any connections to these more standard techniques from the perspective of GFlownets when taking the findings of the paper into account.

**Q9 Complying With Reviewing Instructions:**

Yes

---

> ### Author Rebuttal · Authors · 2024-04-01
>
> We would like to thank the reviewer for their review, and we hope this paper was clear enough despite being outside their field of expertise.
>
> > *The story of the paper seems to be a bit of. The title and abstract do not seem to really match the content of the paper.*
>
> We would like to provide more context for our reasoning behind the title of this paper: we treat **probabilistic inference** over **discrete** objects as learning an optimal policy in MaxEnt RL (**control**). We consider the case where the **environment** is structured in such a way that there are **multiple paths** leading to any (terminating) state.
>
> Moreover, the abstract makes it clear about the content of the paper:
>
> - We treat sampling from a discrete distribution as a sequential decision making problem (Section 2);
> - MaxEnt RL has been used in the past for this, but it is biased for general environments where there are multiple ways of generating the same object (Section 2.2);
> - GFlowNets have been introduced in the past to address this issue (Section 2.3);
> - In this paper, we extend methods to guarantee that we sample from the target distribution with the optimal policy of MaxEnt RL (Section 3.1);
> - We prove equivalences between GFlowNet objectives and MaxEnt RL algorithms (Sections 3.2, 3.3, and Appendix C);
> - We study these equivalences empirically on multiple problems (Section 5).
>
> > *Nevertheless, could the authors comment on more standard inference techniques for discrete random variables. Specifically, are there any connections to these more standard techniques from the perspective of GFlownets when taking the findings of the paper into account.*
>
> Thank you for your suggestion. The main goal of our paper is to build connections between GFlowNets (seen as a method for generative modeling via control) and reinforcement learning methods, similarly to how [Malkin et al., 2023] interprets GFlowNets from the point of view of variational methods, or [Deleu et el., 2023] from the point of view of Markov chains. The connection between GFlowNets & variational methods was mentioned in the extended related work in App. A.1 and will be included in the main paper in the camera ready version. We will also expand this discussion of those connections, showing how GFlowNets unify MaxEnt RL, hierarchical variational inference, and Markov chains.

---

### Official Review · Reviewer_heod · 2024-03-24

**Q2-1 Originality-Novelty:** 3
**Q2-2 Correctness-Technical Quality:** 3
**Q2-5 Clarity Of Writing:** 4

**Q1 Summary And Contributions:**

This paper examines the challenge of sampling of a discrete and structured distribution as a sequential decision problem, aiming to devise a stochastic policy that yields samples at the conclusion of this sequential process in accordance with a predetermined reward scheme. Although Maximum Entropy Reinforcement Learning (MaxEnt RL) could address this for certain distributions, it has been demonstrated that the distribution over states resulting from the optimal policy might exhibit bias when there are multiple paths to generate the same object. On the other hand, Generative Flow Networks (GFlowNets) train a stochastic policy that samples objects in proportion to their reward by approximately maintaining a flow conservation across the entire state space. This paper propose to expand upon recent approaches that adjust the reward to ensure that this goal. The authors claims that certain flow-matching objectives from GFlowNet literature are essentially equivalent to established MaxEnt RL algorithms with a corrected reward. Finally, experiments illustrate the performance of various MaxEnt RL and GFlowNet algorithms across several problems involving sampling from discrete distributions.

**Q2-3 Extent To Which Claims Are Supported By Evidence:**

2: Fair: the main claims are somewhat supported by evidence (but the experimental evaluation may be weak, or does not match entirely with the claims, important baselines may be missing, proofs contain important ideas but lack rigor, algorithmic details are only discussed superficially, references are imprecise, assumptions are not sufficiently motivated or explicated, etc.).

**Q2-4 Reproducibility:**

2: Fair: key resources (e.g. proofs, code, data) are unavailable but key details (e.g. proof sketches, experimental setup) are sufficiently well-described for an expert to confidently reproduce the main results.

**Q3 Main Strengths:**

The propositions, theorems and explanations in this article seem to be very well presented.  This paper has the great merit of providing a clear and easy-to-understand redaction on a rather formal subject. Likewise, the formalization (which is a little difficult because of its mutiplicity depending on the MaxEntRl or GFlowNets model) is perfectly clear.
For me, the main results are to be found more in the equivalences between MaxEntRL and GFlowNets (which will certainly continue to be very fruitful) than in the algorithms described and tested.

**Q4 Main Weakness:**

With the aim of being as complete as possible, the formalization and explanation of equivalences is very long and takes precedence over the more concrete parts. Algorithms and even experimental tasks are too elided and could be described more fully.

The discussion/conclusion is a bit disappointing on the subject of non-deterministic CDMs. Indeed, somehow reciprocally, one could imagine that it is through these equivalences that one could find an unambiguous definition of nondeterministic GFlowNets ?

**Q5 Detailed Comments To The Authors:**

+ Figure 2 needs a lot more explanation.

+ Even if it's quite clear that it's difficult to describe the full details in a 10-page article, it's always annoying to find articles where the size of the supplementary materials is as large as the article itself. For example, A1 and the very relevant Figure 6 (which gives interest to Figure 2) should be in the article itself.

+ A small formalization concern is the status of the final state Sf:

- Sf not in S
- T: SxA->S
and yet : Forall x in X, T(x,a)=Sf

**Q9 Complying With Reviewing Instructions:**

Yes

---

> ### Author Rebuttal · Authors · 2024-04-01
>
> We would like to thank the reviewer for their review and their enthusiasm regarding this paper, as well as their valuable suggestions making the paper stronger. We want to note that if this was a point of concern, all the code is available in the supplementary material for complete reproducibility of our experimental results.
>
> > *The discussion/conclusion is a bit disappointing on the subject of non-deterministic CDMs. Indeed, somehow reciprocally, one could imagine that it is through these equivalences that one could find an unambiguous definition of nondeterministic GFlowNets ?*
>
> There are two attempts in the literature to define GFlowNets in stochastic environments: Stochastic Generative Flow Networks [Pan et al., 2023b] and Expected Flow Networks [Jiralerspong et al., 2024]. The former (Stoch-GFN) does not seem to be a good candidate for an unambiguous definition, as it involves constraints that are not in general satisfiable. The latter (EFN) is similar to but not equivalent to SQL in stochastic environments: under the equivalence of values and log-flows shown in our paper, in EFNs the expectation in the soft Bellman equation is taken in the exponential (flow) domain, rather than in the log (value) domain. We will expand the conclusion section with this discussion.
>
> Going beyond EFNs, which only consider the case where the MDP is a tree, the main technical difficulty of generalizing it to stochastic MDPs with DAG structure is that Theorem 1 relies on the deterministic aspect of the environment to make the telescoping sum of values $V^{*}$ in the proof (App. B) possible.
>
> > *Figure 2 needs a lot more explanation.*
>
> Thank you for the feedback, and we agree that this caption needs more explanation. We will expand the caption of Figure 2 with a text description of the main relationships we prove. We will also attempt to move a condensed version of Figure 6 within the 10 pages of the camera ready version to complement this figure. If this proves to be too challenging given the constraints, we will make sure to reference Figure 6 extensively in the caption of Figure 2.
>
> > *For example, A1 and the very relevant Figure 6 (which gives interest to Figure 2) should be in the article itself.*
>
> We completely agree, and App. A.1 (as well as App. A.2) will definitely be moved back to the main paper in the camera ready version of the paper. As for Figure 6, it is more difficult as it is a full-page figure, but as discussed above we will attempt to include a condensed version of it in the main paper.
>
> > *A small formalization concern is the status of the final state Sf*
>
> This is true, thank you for noting this. The state space of the MDP should be $\bar{\mathcal{S}} = \mathcal{S}\cup \{s_{f}\}$, with the terminal state still distinct from $\mathcal{S}$. With this, the transition matrix should be $T: \mathcal{S}\times \mathcal{A}\rightarrow \bar{\mathcal{S}}$; the rest of the formalism should be valid with this correction.
>
> The motivation for introducing a terminal state which does not belong to $\mathcal{S}$ is because this state is an abstract state which is not designed to ever be a sample of the Gibbs distribution, unlike other states of $\mathcal{S}$ (in some cases, $\mathcal{S}\equiv \mathcal{X}$ being the whole sample space itself, as in Sec. 3.3 & Sec. 5.2).

---

### Official Review · Reviewer_22jf · 2024-03-25

**Q2-1 Originality-Novelty:** 3
**Q2-2 Correctness-Technical Quality:** 2
**Q2-5 Clarity Of Writing:** 3

**Q1 Summary And Contributions:**

The paper addresses the problem of sampling from discrete and structured distributions by treating it as a sequential decision problem.
It presents several theoretical results that show the equivalence between certain GFlowNet objectives and established MaxEnt RL algorithms. The considered methods are experimentally evaluated in three benchmarks.

**Q2-3 Extent To Which Claims Are Supported By Evidence:**

3: Good: the main claims are supported by convincing evidence (in the form of adequate experimental evaluation, proofs, (pseudo-)code, references, assumptions).

**Q2-4 Reproducibility:**

3: Good: key resources (e.g. proofs, code, data) are available and key details (e.g. proofs, experimental setup) are sufficiently well-described for competent researchers to confidently reproduce the main results.

**Q3 Main Strengths:**

- The paper considers an interesting problem.
- The theoretical contribution appears sound, unifying different algorithms.
- The paper is overall clearly written.

**Q4 Main Weakness:**

- It is hard to assess the significance and relevance of the presented results
- Some parts of the paper need clarification (see below)

**Q5 Detailed Comments To The Authors:**

While previous related research consders RL and general MDPs, the methods described are only applicable in very specific settings: deterministic MDPS, the dynamic models are not learned and are quite simple, etc. Actually, the setting seems to fit exactly the framework of linearly-solvable MDPs [Todorov 2009] or optimal control as graphical model inference [Kappen 2012] with a uniform uncontrolled dynamics. I think mentioning that in section 2.1. will add value and clarify the positioning of this contribution.

Can you provide more intuition for Theorem 1 without the need to resort to Lemma 5 of the JMLR paper? Apparently, you are expressing the marginalization over all intermediate steps but the final one, as an expectation over just (one?) trajectory. In that case, P_B(s_t|s_{t+1}) needs to be always one, but that seems to be true in the tree case only, but not in the DAG case.

Also, can you elaborate on how significant is the generalization of Tiapkin et al. [2024] useful? I understand that it extends the applicability of the approach to a more general control problems with intermediate costs. However, it is unclear how relevant is this extension. Can you describe what are the intermediate costs in 5.1, 5.2, and 5.3? Are they non-trivial, i.e., just a constant?

Detailed balance: it appears that detailed balance is required for this to work. Can you confirm that and, if that is the case, what are the implications? How restrictive is this assumption?

**Q9 Complying With Reviewing Instructions:**

Yes

---

> ### Author Rebuttal · Authors · 2024-04-01
>
> We would like to thank the reviewer for their review, and their suggestions which we believe will make our paper stronger.
>
> > *Actually, the setting seems to fit exactly the framework of linearly-solvable MDPs [Todorov 2009] or optimal control as graphical model inference [Kappen 2012] with a uniform uncontrolled dynamics.*
>
> Thank you for your suggestion, and we agree that adding this to the discussion in Sec. 2.1 will make our paper stronger. In fact, it is also interesting to note that the flow-matching condition in (6) is also a linear system in $F$, which connects nicely to linearly-solvable MDPs.
>
> > *Can you provide more intuition for Theorem 1 without the need to resort to Lemma 5 of the JMLR paper?*
>
> The intuition behind Theorem 1 is to transform a *sum* over trajectories into an *expectation* over trajectories (as mentioned in Sec. 3.1).
>
> The terminating state distribution $\pi^{\*}(x)$ is defined in (5) as a sum of $\pi^{\*}(\tau)$ over all trajectories going to $x$, where $\pi^{\*}(\tau)$ is a constant independent of $\tau$ (but depending on $x$) by (4). That’s why the number of trajectories to $x$ appears when the MDP is a DAG (Fig. 1). To remove this effect, it would be convenient to see (5) as an average over trajectories instead of a sum. Therefore, we reweight the terms in the sum (5) with a distribution over trajectories going to $x$. What Lemma 5 of [Bengio et al., 2023] shows is that $P_{B}$ induces such a distribution over trajectories.
>
> > *Also, can you elaborate on how significant is the generalization of Tiapkin et al. [2024] useful?*
>
> The generalization of [Tiapkin et al., 2024] in our Theorem 1 is significant because it allows us to consider cases where intermediate rewards are available, while they only considered the case of a sparse reward obtained only at the end of the trajectory. This in turns allows us to establish 2 connections which were not possible without this generalization: $\pi$-SQL/modified DB (Sec. 3.3), and SQL/forward-looking DB (App. C.4). Moreover, as mentioned in Sec. 3.1, this generalization is applicable to non-Markovian $P_{B}$ used in prior work [Shen et al., 2023].
>
> To further differentiate ourselves from [Tiapkin et al., 2024], we intentionally chose problems where intermediate rewards were available, to highlight the importance of our novel connections.
>
> > *Can you describe what are the intermediate costs in 5.1, 5.2, and 5.3?*
>
> The intermediate rewards are indeed non-trivial:
>
> - Discrete factor graphs (Sec. 5.1): the energy function can be written as a sum of known functions that depend on a subset of variables only (App. D.1). The MDP is structured in such a way that the variables are assigned a value one at a time. The intermediate reward is therefore obtained by evaluating the factors as soon as all the necessary information is available (i.e., all of their inputs have been assigned a value). This reward function is similar to [Buesing et al., 2020], in the case where the MDP is a tree.
> - Bayesian structure learning (Sec. 5.2): all the states are terminating, meaning that all the DAGs $G_{t}$ have a corresponding energy $\mathcal{E}(G_{t})$. We can define the reward as $r(G_{t}, G_{t+1}) = \mathcal{E}(G_{t}) - \mathcal{E}(G_{t+1})$, and with the convention $\mathcal{E}(G_{0}) = 0$, this indeed satisfies $\sum_{t}r(G_{t}, G_{t+1}) = -\mathcal{E}(G_{T})$. In the context of structure learning, this is called the delta-score, and [Deleu et al., 2022] have shown that it can be computed efficiently.
> - Phylogenetic tree generation (Sec. 5.3): the energy function is the total number of mutations encoded in a phylogenetic tree. The total number of mutations can be decomposed as the sum of (1) the number of mutations at the root of the tree, and (2) the total number of mutations in the left & right subtrees. We can use the number of mutations at the root as the intermediate reward, which is computed using the Fitch algorithm [Zhou et al., 2024].
>
> > *Detailed balance: it appears that detailed balance is required for this to work. Can you confirm that and, if that is the case, what are the implications?*
>
> "Detailed balance" here relates to the detailed balance condition from the GFlowNet literature, and is not exactly the detailed balance in the Markov chain literature (although [Bengio et al., 2023] reference it as a source of inspiration). The DB condition in GFlowNets, leading to the DB loss in (38) App. C.2, is a condition which leads to a policy whose terminating state distribution matches the Gibbs distribution. Therefore this is a desired property that is being learned with the DB loss, rather than a requirement.
>
> Despite connections between GFlowNets & Markov chains [Deleu et al., 2023], the main difference is that the reversibility of the Markov kernel is often assumed in Markov chains (we can move $s\rightarrow s'$ & $s'\rightarrow s$), notably when talking about detailed balance, which is not the case for GFlowNets since the MDP has a DAG structure.

---

### Meta-Review · Area_Chair_JeSq · 2024-04-14

This paper addresses the problem of sampling from discrete and structured distributions, from a perspective of sequential decision-making. It presents several theoretical results that show the equivalence between certain GFlowNet objectives and established MaxEnt RL algorithms, followed by thorough experimental results. The paper is well-written and well-motivated, with novel perspectives. There were some comments regarding the clarity of the paper, and I encourage the authors to address them in preparing the next version of the paper.